# A newly integrated ground temperature dataset of permafrost along the China-Russia Crude Oil Pipeline route in Northeast China

Guoyu Li[1,3,4], Wei Ma[1,3,4], Fei Wang[1, 2*], Huijun Jin[1, 3, 5*], Alexander Fedorov[6], Dun Chen[1,3], Gang Wu[1,3,4], Yapeng Cao[1,3,4], Yu Zhou[1,3,4], Yanhu Mu[1,3,4], Yuncheng Mao[7], Jun Zhang[8], Kai Gao[1,3,4], Xiaoying Jin[5], Ruixia He[1,3], Xinyu Li[9], and Yan Li[1,3]

[1] State Key Laboratory of Frozen Soil Engineering, Northwest Institute of Eco-Environment and Resources, Chinese Academy of Sciences, Lanzhou 730000, China;

[2] Faculty of Civil Engineering and Mechanics, Jiangsu University, Zhenjiang 212013, China;

[3] Da Xing'anling Observation and Research Station of Frozen-Ground Engineering and Environment, Northwest Institute of Eco-Environment and Resources, Chinese Academy of Sciences, Jagdaqi, Inner Mongolia 165000, China;

[4] School of Engineering Science, University of Chinese Academy of Sciences, Beijing 100049, China;

[5] School of Civil Engineering and Permafrost Institute, Northeast Forestry University, Harbin 150040, China;

[6] Melnikov Permafrost Institute, Russian Academy of Sciences, Yakutsk 677010, Russia;

[7] School of Civil Engineering, Northwest Minzu University, Lanzhou 730000, China;

[8] School of Civil Engineering, Henan Polytechnic University, Jiaozuo 454000, China, and;

[9] School of Civil Engineering, Harbin Institute of Technology, Harbin 150090, China

Correspondence: Fei Wang (wangfei9107@ujs.edu.cn) and Huijun Jin (hjjin@nefu.edu.cn)

**Abstract:** Thermal state of permafrost in the present and future is fundamental to ecosystem evolution, hydrological processes, carbon release, and infrastructure integrity in cold regions. In 2011 we initiated a permafrost monitoring network along the China-Russia Crude Oil Pipelines (CRCOPs) route at the eastern flank of the northern Da Xing'anling Mountains in Northeast China. We compiled an integrated dataset of the ground thermal state along the CRCOPs route, consisting of meteorological data near the southern limit of latitudinal permafrost, ground temperature data in 20 boreholes with depths of 10.0–60.6 m, soil volumetric liquid water contents and 2-dimensional electrical resistivity tomography (ERT) data at different sites. Results demonstrate a permafrost warming during 2011–2020 in the vicinity of the southern limit of latitudinal permafrost, as manifested by rising ground temperatures at almost all depths in response to climate warming. Local thermal disturbances triggered by the construction and operation of CRCOPs have resulted in significant permafrost warming and subsequent thawing on the right-of-way (ROW) of the pipelines. This permafrost thaw will persist, but it can be alleviated by adopting mitigative measures, such as an insulation layer and thermosyphons. The *in-situ* observational dataset is of great value for assessing the variability of permafrost under the linear disturbances of the CRCOPs and related environmental effects, for understanding hydro-thermal-mechanical interactions between the

buried pipelines and permafrost foundation soils, and for evaluating the operational and structural integrity of the

pipeline systems in the future. The dataset is available at the National Tibetan Plateau/Third Pole Environment Data

Center (http://doi.org/10.11888/Cryos.tpdc.272357; Li, 2022).

# 1 Introduction

As a major component of the Earth's cryosphere, permafrost is sensitive to climate change, surface disturbances and human activities (Smith et al., 2022). Over the last few decades, the warming and thawing of permafrost have been observed in most permafrost regions (e.g., Ran et al., 2018; Biskaborn et al., 2019; O'Neill et al., 2019; Etzelmüller et al., 2020; Liu et al., 2021; Noetzli et al., 2021; Smith et al., 2022), and permafrost degradation will continue in response to a warming climate (Koven et al., 2013; Burke et al., 2020). Permafrost degradation affects the geomorphological characteristics, carbon release, hydrological process, ecosystem, climate system, and integrity of infrastructure (Cheng and Jin, 2013; Beck et al., 2015; Hjort et al., 2018, 2022; Turetsky et al., 2020; Jin and Ma, 2021; Jin et al., 2021, 2022; Luo et al., 2021; Jones et al., 2022; Liu et al., 2022; Miner et al., 2022).

Permafrost occurs extensively in the Da and Xiao Xing'anling mountains in Northeast China (referred to as the Xing'an permafrost). The estimated areal extent of existing permafrost in Northeast China ranges from $2.4\times10^5$ to $3.1\times10^5 \, km^2$ (Ran et al., 2012; Zhang et al., 2021). Its distribution displays both latitudinal and altitudinal zonality and is strongly influenced by local environmental factors (Jin et al., 2008; Guo et al., 2018; He et al., 2021). The Xing'an permafrost has also experienced significant degradation under a warming climate, wildfires, and human activities, such as deforestation, urbanization, mining and linear infrastructure construction (Guo and Li, 1981; Jin et al., 2007; Wang et al., 2019a; Mao et al., 2019; Li et al., 2021; Serban et al., 2021), as evidenced by rising ground temperature (GT), thickening active layer, development of taliks, shrinking permafrost extent and increases in thaw-related landscape change and hazards such as ground surface subsidence, settlement of foundation soils and thermokarst. Multiple studies on future changes in Xing'an permafrost have been conducted based on different modeling approaches and climate warming scenarios (e.g., Ran et al., 2012; Zhang et al., 2021). Research results indicate that persistent permafrost degradation is likely to occur during the next few decades (Jin et al., 2007; Wei et al., 2011). However, there are great uncertainties in the prediction of the magnitude and timing of these changes

(Smith et al., 2022). Field observations of meteorological variables and permafrost thermal states have substantially
contributed to the understanding of the responses of GTs to climate change and to hydrothermal processes in the
active layer and permafrost, facilitating the evaluation and/or validation of predictive permafrost models, and thus
they are of great importance (Zhao et al., 2021; Wu et al., 2022). However, in Northeast China, long-term and
continuous datasets of permafrost thermal state are scarce, especially at the eastern flank of the Da Xing'anling
Mountains, due to the harsh periglacial environment, inconvenient access, and expensive installation and
maintenance costs (Jin et al., 2007; He et al., 2021; Li et al., 2021).

Since 2008, extensive permafrost investigations for the construction of the China-Russia Crude Oil Pipelines

(CRCOPs) I and II were conducted in the permafrost zones on the eastern slopes of the Da Xing'anling Mountains.
The CRCOP I was constructed during the two cold seasons in 2009-2010 and began operation in January 2011. In
January 2018, CRCOP II was completed and began to operate. A permafrost monitoring network along the
CRCOPs route was gradually established by referring to the experiences and lessons learned from other oil and gas
pipelines (e.g., Norman Wells to Zama crude oil pipeline in Canada, Alyeska crude oil pipeline in the U.S., and
Nadym–Pur–Taz natural gas pipeline in Russia) in permafrost regions (Seligman, 2000; Burgess and Smith, 2003;
Johnson and Hegdal, 2008; Smith and Riseborough, 2010; Oswell, 2011). Boreholes were instrumented to measure
GTs in the active layer and near-surface permafrost on and off the right-of-way (ROW) of the CRCOPs and
electrical resistivity tomography (ERT) surveys were used to delineate frozen and unfrozen ground in the vicinity of
the CRCOPs (Kneisel et al., 2008; Farzamian et al., 2020).

We firstly present the integrated dataset of permafrost thermal state along the CRCOPs route on the eastern

slopes of the northern Da Xing'anling Mountains in Northeast China. This dataset includes meteorological data,
GTs, soil volumetric liquid water content, and subsurface electrical resistivity (ER) on and off the ROW of the
pipeline. Detailed information for the integrated dataset is provided so that this dataset can be easily understood,
readily accessed and properly applied by potential users.

## 2 Site description

Five permafrost observation sites, named as Xing'an (XA), Xin-tian (XT), Jin-song (JS), Song-ling (SL), and Jagdaqi Bei (North) (JB), respectively, were established along the CRCOPs route in Northeast China (50.4710°–53.3328°N, 123.9875°–124.3132°E) (Fig.1) through the joint efforts of the State Key Laboratory of Frozen Soil Engineering (SKLFSE), Northwest Institute of Eco-Environment and Resources (the former Cold and Arid Regions Environmental and Engineering Research Institute), Chinese Academy of Sciences and the Jagdaqi Division of the PetroChina Pipeline Company. Site selection was primarily based on engineering geological conditions of permafrost (Jin et al., 2010). According to the meteorological data of 1972–2017, the study area is characterized of a frigid-temperate continental monsoon climate with mean annual air temperatures (MAAT) of −4.0 to −0.4 °C, with annual precipitation of 447 to 525 mm, which falls mostly as summer rain. Snow cover generally occurs at the end of September–beginning of October and disappears in late April and early May of next year. The snow depth ranges from 5 to 35 cm. Between 1972 and 2017, MAAT increased at a rate of 0.32 °C per decade while annual precipitation increased at a rate of 14.6 mm per decade (Wang et al., 2019a).

Table 1 summarizes the geographical information and permafrost characteristics of monitoring sites. Permafrost is warm with mean annual ground temperature (MAGT) at the depth of zero annual amplitude ($D_{ZAA}$) ranging from −1.8 to −0.4 °C. The permafrost thickness varies from 0 to more than 60 m, and the observed active layer thickness (ALT) ranges from 1.0 to 2.7 m (Wang et al., 2019b). Along the CRCOP route, the XA site, located in a permafrost wetland, is the most northern and has the lowest air temperature, while the JB site, near the southern limit of the latitudinal permafrost in Northeast China, has the highest air temperature, and permafrost occurs in isolated patches. The XT and JS sites are located in the transition zone between isolated patches of permafrost and sporadic permafrost, making them the ideal locations for examining permafrost dynamics. The SL site is located in a wetland underlain by ice-rich permafrost, where seasonal frost mounds, sometimes migratory, with a maximum height of 2 m are developed (Wang et al., 2015), and monitoring devices are prone to be destroyed due to

significant frost mound-related ground deformation.

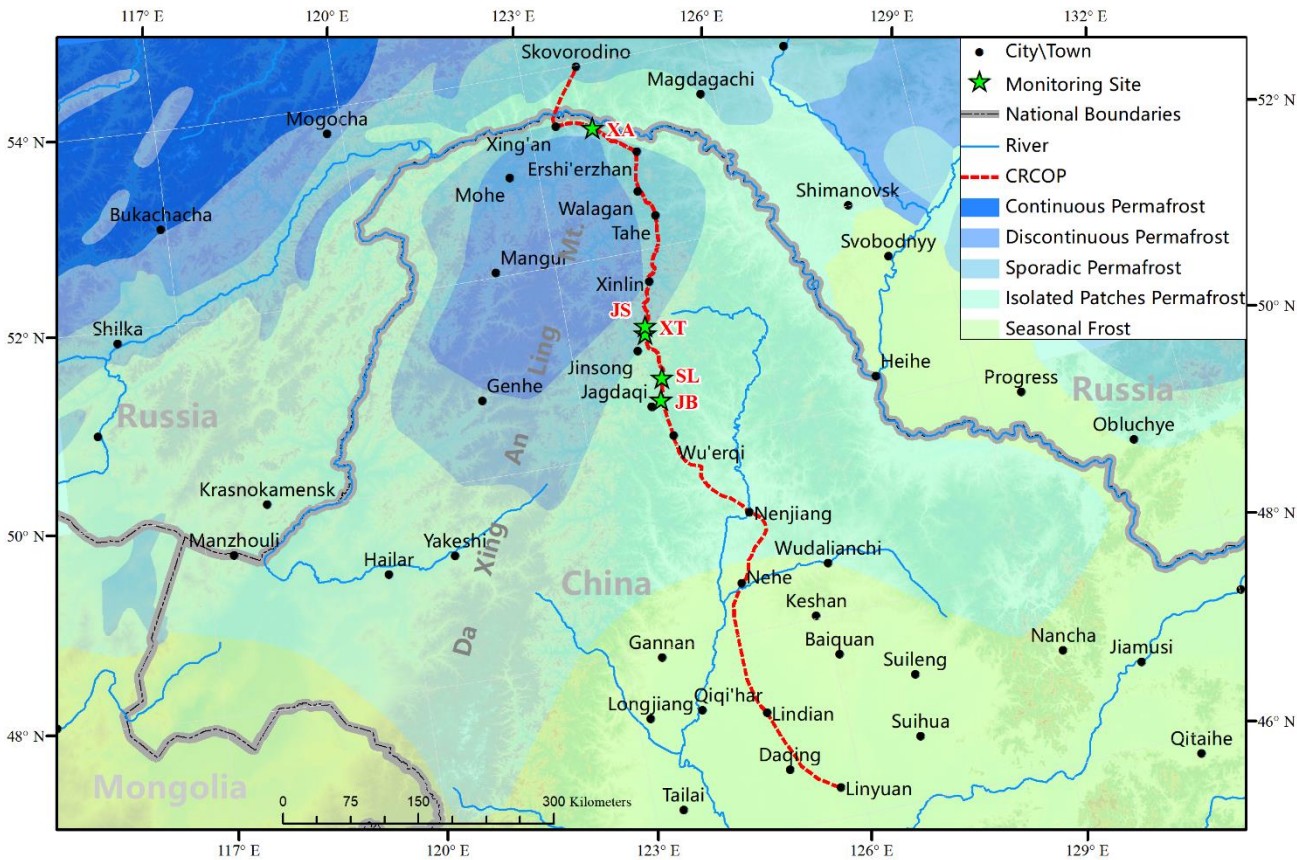

Figure 1. Location of permafrost monitoring sites along the route of China-Russia Crude Oil Pipelines (CRCOPs).
Permafrost zone from Jin et al., (2007, 2010). The red dash line represents the paralleling CRCOPs I and II (the inter-
pipeline distance is generally limited to approximately 10 m).

Table 1. Permafrost characteristics for monitoring sites along the route of China-Russia Crude Oil Pipelines.

| Site | Lat. (°N) | Long. (°E) | Elev. (m a.s.l.) | Permafrost zone | Vegetation | Ice content | MAGT (°C) | ALT (cm) |
|------|-----------|-----------|------------------|-----------------|------------|-------------|-----------|----------|
| XA | 53.3328 | 123.9875 | 318 | Sporadic permafrost | *Carex tato swamp* | Ice-saturated | −1.8 | 130 |
| XT | 51.2444 | 124.2096 | 621 | Sporadic permafrost | *Shrub meadow* | Ice-saturated | −1.8 | 100 |
| JS | 51.1619 | 124.1943 | 508 | Sporadic permafrost | *Carex tato swamp* | Ice-rich | −0.7 to −0.4 | 200~270 |
| SL | 50.6868 | 124.3132 | 398 | Isolated patches permafrost | *Carex tato swamp* | Ice-saturated /ice-rich | −0.9 | 130 |
| JB | 50.4710 | 124.2257 | 484 | Isolated patches permafrost | *Carex tato swamp* | Ice-rich | −0.8 to −0.5 | 178~200 |

Notes: MAGT, mean annual ground temperature of permafrost at the depth of zero annual amplitude, and ALT,
active layer thickness.

# 3 Data description

## 3.1 Meteorological data

In October 2017, a small automated weather station (AWS, Fig. B1) was installed at the JB site for measuring air temperature, relative humidity, wind speed and direction, and total solar radiation. Details of the sensors used are listed in Table 2. All meteorological data were recorded every two hours by a CR3000 data logger (Campbell Scientific, Inc., USA) with a relay multiplexer (TRM128, China), powered by a battery cell and solar panel regulated by a solar charge controller (Phocos ECO (10 A), Germany). The collected data have been transferred automatically to the specified server by the wireless transmission module (HKT-DTU, Campbell Scientific, Inc., USA). Using such technology, it would be possible to check collected data in real-time and identify possible sensor failures.

Table 2 List of sensors, measuring range and accuracy for meteorological data, ground temperature, soil water content, and ground electrical resistivity.

| Variable | Sensor/measurement device | Measuring range (operating temperature) | Accuracy |
|---|---|---|---|
| Meteorological data | | | |
| Air temperature | HMP155A Vaisala Finland | −80 to 60 °C | $(0.226-0.0028 \times T)$°C $(-80$ to $20$°C), $(0.055+0.0057 \times T)$°C $(20$ to $60$ °C) |
| Relative humidity (RH) | HMP155A Vaisala Finland | 0 to 100% RH | $(1.4+0.032 \times RH)$% $(-60$ to $-40$°C), $(1.2+0.012 \times RH)$% $(-40$ to $-20$°C, $40$ to $60$°C), $(1.0+0.008 \times RH)$% $(-20$ to $40$°C) |
| Wind speed/direction | Model 05103 R.M. Young Company | 0 to 100 m/s (−50 to 50°C) | $\pm 0.3$ m s$^{-1}$, $\pm 3°$ |
| Total solar radiation | LI200X Pyranometer Campbell Scientific, Inc. | 0~1000 W/m$^2$ (−40 to 65°C) | $\pm 5$% (absolute error in natural daylight), $\pm 3$% typical |
| Permafrost monitoring | | | |
| Soil temperature | Thermistor cable SKLFSE, China | −30 to 30 °C | $\pm 0.05$°C |
| Soil volumetric liquid water content | CS616 Campbell Scientific, Inc. | 0% to saturation (0 to 70°C) | $\pm 2.5$% |
| Ground electrical resistivity | SuperSting R8 system Advanced Geosciences, Inc. | −10 to 10 V | $\leq$30nV |

The AWS was regularly maintained and repaired, resulting in data collection with satisfactory quality and continuity. Between 15 October 2017 and 10 August 2020, less than 5% of the data were missing. However, the meteorological data had been discontinued since 10 August 2020 due to the failure of the online data transmission

module and lack of essential on-site maintenance for equipment under the influence of the COVID-19 pandemic.
Air temperature and relative humidity were measured at a height of 1.5 m every two hours using the Vaisala
HMP155A sensor protected by a radiation shield. The accuracy of temperature and relative humidity measurements
decreased along with lowering temperatures. For example, the accuracy for the HMP155A sensor was as good as
±0.17 °C at an ambient temperature of 20 °C, but only ± 0.34 °C at −40 °C. At the JB permafrost site, the annual
range of daily air temperature was approximately 56 °C. The recorded maximum air temperature was 24.7 °C on 25
July 2020, and the minimum, −33.7 °C on 27 December 2019 (Fig. 2a). The seasonal variation in relative humidity
followed similar patterns with the seasonal variability in air temperature (Fig. 2b).

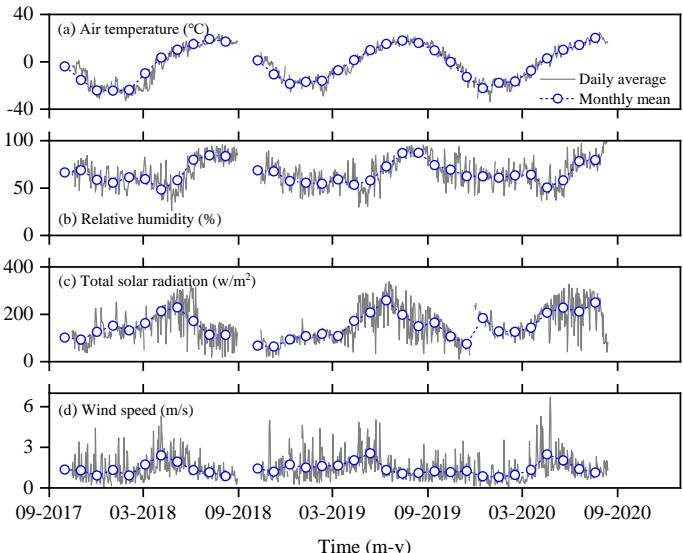


Figure 2. Time series of meteorological variables at the JB site from October 2017 to August 2020.

Total solar radiation was measured at a height of 1.5 m above the ground surface by the LI200X Pyranometer
with an accuracy of ±3% to ±5%). Although the sensors were regularly checked (e.g., checking the level of the
instrument and removing dust) during the site visits and re-calibrated after two years of installation, the instruments
were largely unattended and their accuracy was therefore likely to worsen up to ±5%. It is worth noting that the
LI200X may read negative solar radiation during the night, which is later set to zero in the data processing. The
total solar radiation reached its maximum in summer (June to August) and was lowest in winter (December to next
February), with a similar pattern with seasonal variations in air temperature (Fig. 2c).

The wind speed and direction were measured at a height of 2 m above the ground surface by a propeller

anemometer (Model 05103, R.M. Young Company). The standard error of wind direction was also calculated using

a specific algorithm provided by the CR3000 data logger. The recorded maximum wind speed of 9 m s$^{-1}$ occurred

on 28 May 2019. The average monthly wind speeds ranged from 0.9 to 2.6 m s$^{-1}$ and reached their maximum in

April–May (Fig. 2d).

**3.2 Ground temperature and soil water content data**
In total, 20 boreholes were drilled and instrumented for GT monitoring both on the ROW of the CRCOPs at
varying distances from the pipe centerline/axis-lines and in nearby undisturbed terrains (off the ROW, 2.6 to 90 m
from the ROW edge) between 2011 and 2021 (Table 3). Generally, the ROW is approximately 20 m wide. The
depths of boreholes range from 10.0 to 60.6 m, although most of them are 15 to 20 m deep. *In-situ* temperature
monitoring in the borehole JB-B-II (6.6 m from the ROW edge) was deployed starting in November 2011, and ten
boreholes (JB-B-1 to JB-B-10) were drilled on the ROW between 2 and 4 m from the centerline of pipe in 2014 and
2015 at the JB site. Besides, a new borehole (JB-B-I) was drilled down to 60.6 m near the above-mentioned AWS in
March 2017. At the SL site, two on-ROW boreholes (SL-B-1 and SL-B-2, 3 and 5.9 m from the centerline of
CRCOP II, respectively) and one off-ROW borehole (SL-B-I, 2.6 m from the edge of the CRCOP I ROW) were
drilled in March 2017 and instrumented in September 2017. At the JS site, two boreholes (JS-B-1 and JS-B-2, 2 and
5 m from the pipe centreline, respectively) were established on the CRCOP II ROW in 2017, and an additional
borehole (JS-B-I) was drilled 14.8 m from the ROW edge of CRCOP I in 2018. A borehole (XT-B-I) at the XT site,
10 km north of the JS site, was drilled in 2019 to evaluate the spatial differentiation of permafrost distribution
influenced by local geo-environmental factors. At the XA site, a borehole with a depth of 60.5 m (XA-B-I) was
drilled 7 km south of the first pump station of the CRCOPs in Xing'an Village of Mo'he County, Heilongjiang
Province, where there was previously no GT data.

Table 3 Summary of monitoring information of ground temperature boreholes, water content monitoring pits, and electrical resistivity tomography (ERT) profiles along the China-Russia Crude Oil Pipelines (CRCOPs) in Northeast China.

| Variable | Borehole/ ERT profile | Maximum monitoring depth (m) | Distance from pipe centreline (m) | Data logger | Measuring internal | Operation period |
|---|---|---|---|---|---|---|
| Soil/permafrost temperature at the natural site (off-ROW) | XA-B-I | 60.5 | 100 | CR 3000 | 2h, AUTO | Nov 2018 – Nov 2020 |
| | XT-B-I | 20 | 80 | RTB37a36V3 | 2h, AUTO | Jul 2019 – Aug 2021 |
| | JS-B-I | 20 | 24.8 | CR 3000 | 2h, AUTO | Dec 2018 – Jun 2021 |
| | SL-B-I | 25 | 12.6 | Fluke 87/89 RTB37a36V3 | Monthly, MANU 2h, AUTO | Sep 2017 – Oct 2019 Aug 2020 – Dec 2020 |
| | JB-B-I | 60.6 | 80 | CR 3000 | 2h, AUTO | Jun 2018 – Aug 2020 |
| | JB-B-II | 20 | 16.6 | Fluke 87/89 CR 3000 | Monthly, MANU 2h, AUTO | Nov 2011 – Sep 2017 Oct 2017 – Aug 2021 |
| Soil/permafrost temperature on pipeline ROW (on-ROW) | JS-B-1* | 19.8 | 2 | CR 3000 | 2h, AUTO | Oct 2017 – May 2021 |
| | JS-B-2* | 20 | 5 | CR 3000 | 2h, AUTO | Oct 2017 – Aug 2021 |
| | SL-B-1* | 24.8 | 3 | Fluke 87/89 RTB37a36V3 | Monthly, MANU 2h, AUTO | Sep 2017 – Oct 2019 Aug 2020 – May 2021 |
| | SL-B-2* | 24.8 | 5.9 | Fluke 87/89 RTB37a36V3 | Monthly, MANU 2h, AUTO | Sep 2017 – Oct 2019 Aug 2020 – May 2021 |
| | JB-B-1 | 20 | 2 | Fluke 87/89 CR 3000 | Monthly, MANU 2h, AUTO | Mar 2014 – Sep 2017 Oct 2017 – Aug 2021 |
| | JB-B-2 | 15 | 2 | CR 3000 | 2h, AUTO | Jun 2015 – Aug 2021 |
| | JB-B-3 | 15 | 2 | | 2h, AUTO | Jun 2015 – May 2018 |
| | JB-B-4 | 15 | 2 | | 2h, AUTO | Jun 2015 – May 2019 |
| | JB-B-5 | 10 | 3 | | 2h, AUTO | Jun 2015 – Aug 2021 |
| | JB-B-6 | 14 | 3 | | 2h, AUTO | Jun 2015 – May 2018 |
| | JB-B-7 | 15 | 3 | | 2h, AUTO | Jun 2015 – May 2020 |
| | JB-B-8 | 15 | 4 | | 2h, AUTO | Jun 2015 – Aug 2021 |
| | JB-B-9 | 15 | 4 | | 2h, AUTO | Jun 2015 – May 2018 |
| | JB-B-10 | 15 | 4 | | 2h, AUTO | Jun 2015 – May 2020 |
| Soil volumetric liquid water content on pipeline ROW | JB-W1 | 2.5 | 1 | CR 3000 | 2h, AUTO | Jun 2015–Aug 2021 |
| | JB-W2 | | | | | |
| | JB-W3 | | | | | |
| Electrical resistivity | P-JS | 24 | | SuperSting R8 system | Site visit | Apr 11, 2018 |
| | P-SL | 24 | | | Site visit | Apr 12, 2018 |
| | P-JB-1 | 24 | | | Site visit | Apr 06 – Apr 10, 2018 |
| | P-JB-2 | 24 | | | | |
| | P-JB-3 | 24 | | | | |
| | P-JB-4 | 18 | | | | |

* Boreholes were drilled on the ROW of CRCOP II. The width of ROW along the pipeline is about 20 m.

The GT measurement was carried out by installing a thermistor cable protected by a steel tube into the
borehole (Wang et al., 2019b). The thermistor cable was assembled by the SKLFSE, with thermistors at the
designed intervals. Manual temperature reading using Fluke 87/89 was made in five boreholes (SL-B-I, SL-B-1,
SL-B-2, JB-B-II, and JB-B-1) for some earlier time (Table 3). The accuracy of the manual readings is estimated to
be ±0.1 °C (Juliussen et al., 2010). Two types of data loggers, which are connected to the thermistor cables, are now
used for automatic and continuous GTs monitoring in boreholes at 2 h intervals. They are CR3000 data loggers and
miniature temperature data loggers (RTB37a36V3, jointly developed by Northwestern Polytechnical University and
SKLFSE). The latter generally has a lower resolution than the CR3000 data logger (±0.05°C), but allows more
widely used due to its lower cost. The GT in boreholes at the SL site has been recorded by this miniature data
logger. The soil volumetric liquid water content (VWC) was measured by the Campbell Scientific CS616 water
content reflectometer probe (Table 2) by connecting to a CR3000 data logger. Three pits were excavated on the
ROW of the CRCOP I at the JB site and three probes were embedded horizontally at depths of 0.5, 1.5, and 2.5 m
in each pit (Table 3).
Quality control of data was carried out by manually checking to detect missing data and obvious erroneous
recordings in the GT and VWC data. All the missing or abnormal data were replaced with null values. Then, daily
averages were calculated from hourly values at 2-hour intervals if at least 10 values (~ 83 %) were available within
1 day.
**3.2.1 Ground temperature at the undisturbed sites**
To analyze the spatial distribution of GT, we chose GTs between 2018 and 2021, when GT data series of all five
permafrost sites were available (Fig. 3). The average daily GTs at depths of 0–3 m showed seasonal variations, but
the amplitude decreased with depth, with the magnitude of the decrease varying between sites (Fig. 3a). For
example, the JS site with a high permafrost temperature had the maximum annual range of GT at the depth of 0.5 m
(from −14.9 to 16.6 °C) among all the five sites, while at the depth of 3 m, the XT site had the maximum range in
GT, which was mainly related to the local topography, vegetation, soils and geology (Table 1). Zero-curtain effects
were evident at a certain depth in these five sites, but the duration time of zero curtains at the same depth varied
greatly with location, which was mainly related to *in situ* soil water/ice content of these permafrost sites.

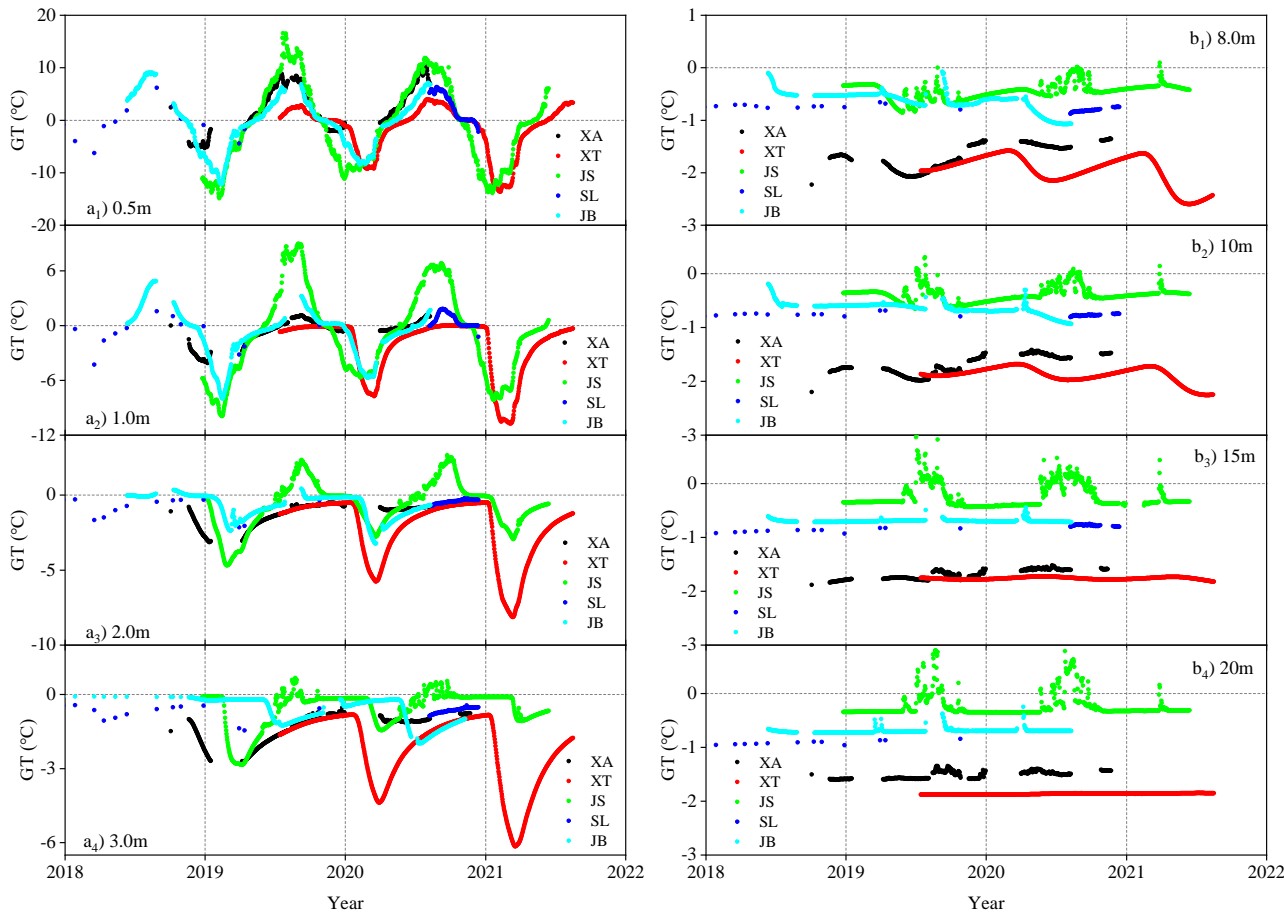


Figure 3. Variability of ground temperatures at depths of 0–3 m (a) and 8–20 m (b) at the undisturbed sites along the
route of China-Russia Crude Oil Pipelines (CRCOPs) in Northeast China, 2018–2021.
Seasonal variations in GTs at depths $\geq$ 15 m are negligible at all sites, except for the JS site, which indicates
that zero annual amplitude (ZAA) is located below 15 m in depth (Fig.3b). At the JS site, abnormal positive
temperatures were observed in the summers of 2019 and 2020, probably due to the thermal disturbance of supra-
and/or intra-permafrost groundwater. GTs at depths of 8, 10, 15, and 20 m showed that permafrost temperature
decreased northward. However, there is substantial scatter in the relationship between GT and latitude (Fig. 4).

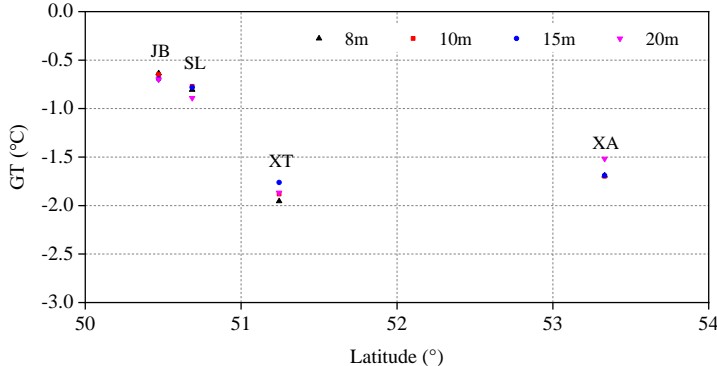


Figure 4. Relationship between latitude and GT along the route of China-Russia Crude Oil Pipelines (CRCOPs).

A decade record (2011–2020) of GTs in the active layer and near-surface permafrost in borehole JB-B-II
allows for assessment of the inter-annual trend of permafrost. As displayed in Fig. 5, the monthly average GTs in
2018 at depths from 1 to 2 m were fluctuating in proximity to 0 °C without an obvious geothermal gradient (termed
as the zero curtain layer), decreased with a geothermal gradient of 0.08 °C m$^{-1}$ at depths from 2 to 7 m, and
remained unchanged below 7 m. The ALT (the maximum depth of 0 °C isotherm from linear interpolation of the
daily average GTs) in this borehole varied slightly between 178 and 200 cm from 2011 to 2020, mainly due to the
damping effect of the zero curtain layer and thermal properties of soil deposits (Fig. 6a), while the near-surface
permafrost at depths of 8-20 m was warming at an average rate of 0.035 °C yr$^{-1}$ in this 10-year observation period
(Fig. 6b). At the $D_{ZZA}$ of 15 m, MAGT increased by 0.3 °C (from −0.8 to −0.5 °C) during 2011-2020 (Romanovsky
et al., 2010; Smith et al., 2010).

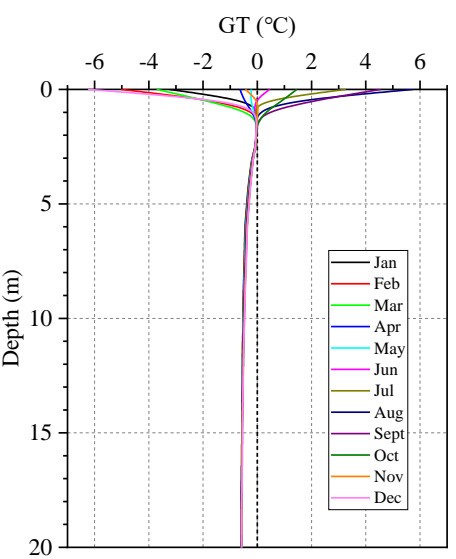


Figure 5. Monthly average ground temperatures at depths of 0-20 m recorded in the JB-B-II borehole at the JB site in 2018.

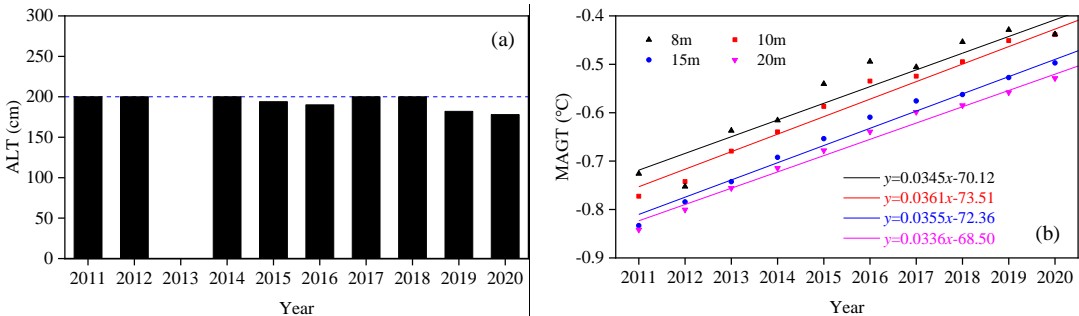

Figure 6. Variations in the active layer thickness (ALT) (a) and mean annual ground temperature (MAGT) (b) in borehole JB-B-II at the JB site along the China-Russia Crude Oil Pipeline (CRCOP) I in northern part of Northeast China from 2011 to 2020.

**3.2.2 Ground temperature on the ROW of pipeline**

Boreholes on the ROW along the CRCOPs were drilled and instrumented for GT monitoring at three permafrost sites (JS, SL, and JB) to evaluate the thermal disturbances of the insulated or uninsulated CRCOPs on the surrounding permafrost (Table 3). At the JS and SL sites, the pipelines were insulated and initially buried at depths of 2-3 m, while they were uninsulated and buried at a depth of 1.6-2.4 m at the JB site. The warm oil (with the maximum recorded temperature of 28 °C at the first/northernmost pump station of the CRCOPs in China) flowing in the pipeline brought substantial heat into the underlying and permafrost, resulting in the rising GTs, even though the pipelines were wrapped by an insulation layer (Fig. 7). However, temperature differences on and off the ROW of pipeline were substantially reduced by an insulation layer at the JS and SL sites compared to those at the JB site without insulation. GTs in the borehole 5 m (perpendicular to the CRCOP I) away from the pipe centerline were also greater than those at the nearby undisturbed site, indicating the lateral thermal disturbance range of the pipeline may have extended beyond a horizontal distance of 5 m.

The time-series of the depths of the permafrost table and maximum frost penetration in borehole JB-B-1, horizontally 2 m away from the uninsulated pipe centerline at the JB site shows that since the initiation of CRCOP 1 operation in 2011, the depth of the permafrost table has been increasing with an average rate of 0.9 m yr$^{-1}$ and depth of seasonal frost penetration decrease rapidly and then varies little (0.4-0.8 m) during 2010–2021 (Fig. 8).

Therefore, a thawed interlayer between permafrost table and the bottom of seasonal frost (i.e., supra-permafrost
subaerial talik, SST) formed and developed with an average rate of 1.1 m yr$^{-1}$ in the same period. This has
demonstrated that the pipeline has triggered an intensive and quick permafrost degradation at a local scale. This
deepening of the permafrost table and thickening of the SST have led to significant subsidence of the ground
surface within the trench area and exposed the pipelines to thawed low-bearing foundation soils, resulting in
potential pipeline damage. For example, the excavation at the JB site in 2015 revealed that the CRCOP I had locally
settled down by 1.4 m during 2010–2015.

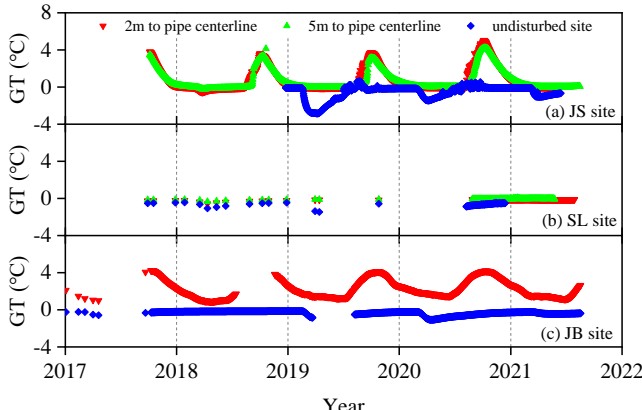

Figure 7. Variations in ground temperatures at the depth of 3 m on the right-of-way (ROW) at the JS (a), SL (b), and JB
(c) sites along the China-Russia Crude Oil Pipeline (CRCOP) I in northern part of Northeast China during 2017–2021.

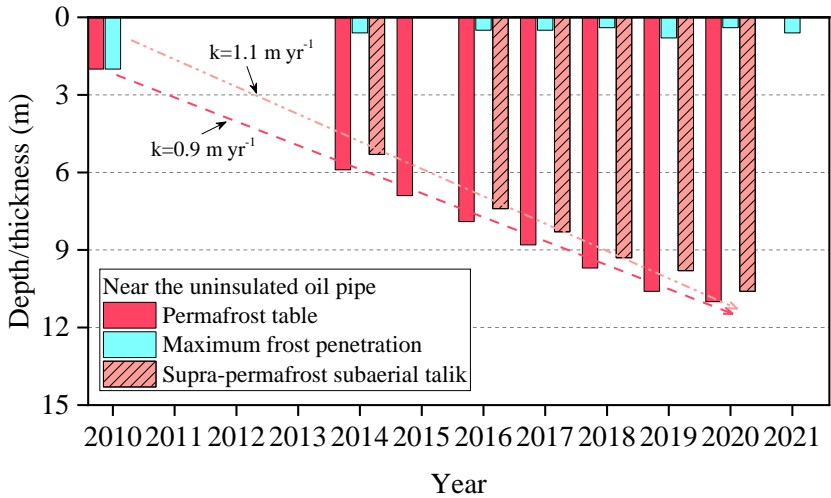

Figure 8. Variations in the depths of permafrost table and seasonal frost, and thickness of supra-permafrost subaerial talik
during 2010–2021 in borehole JB-B-1, 2 m away from the centerline of the uninsulated China-Russia Crude Oil Pipeline
(CRCOP) I at the JB site in northern part of Northeast China. Where k denotes the increasing rate.
Thermosyphons, a widely-used mitigative measure for permafrost thaw in cold region engineering, can
effectively change the temporal and spatial variations of local GT distribution. Figure 9 shows the time series of
temperature contours in boreholes of JB-B-2, JB-B-3, JB-B-6, and JB-B-9 from July 2015 to August 2021 (the
period with high-quality GT data series). Borehole JB-B-2 is located 2 m away from the pipeline centerline with
one pair of thermosyphons, while boreholes JB-B-3, JB-B-6, and JB-B-9 are set 2, 3, and 4 m, respectively, away
from the centerline of the CRCOP I, but with two pairs of thermosyphons. During the cold season, the permafrost
table remains unchanged due to the cooling effect of thermosyphons, but it deepens when the thermosyphons stop
working during the warm season. Overall, the depth of the permafrost table has been increasing slowly over the
observational decade (Fig.9a). The cooling performance of the thermosyphons on pipeline foundation soils has
been enhanced with an increased number of thermosyphons, as characterized by a lower rate of permafrost table
deepening and a wider vertical cooling extent in winter or the cold season (Fig.9b). GTs in boreholes of JB-B-6 and
JB-B-9 indicate a greater-than-1.5-m cooling range of thermosyphons, and a greater-than-4-m lateral extent on the
ground surface of thermal disturbance of the warm pipeline (CRCOP I) (Figs.9c and 9d). During June–August
2020, abnormal changes in 0 °C isotherm shown in Figures 9b, 9c, and 9d are likely related to the infiltration of
surface waters and supra- and/or intra-permafrost waters. The above results show that the vertically-inserted
thermosyphons are unable to completely prevent the thawing of the underlying permafrost. The unexpectedly
warmer oil temperature, thermal erosion of surface water ponding on the ROW, and lowering thermosyphon
performance are responsible for the unsatisfactory cooling effect of thermosyphons on the pipeline foundation soils.

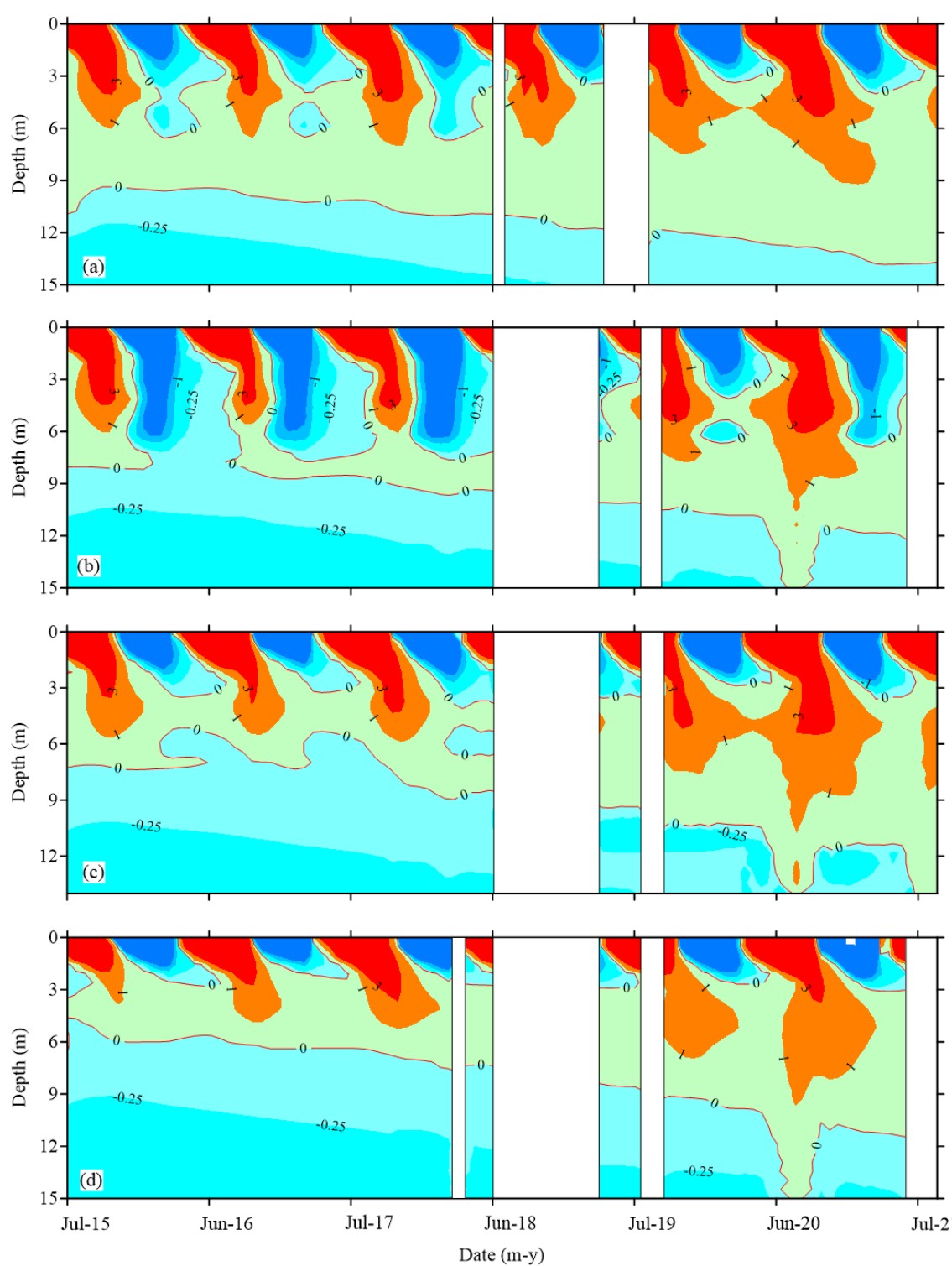

Figure 9. Depth-time contour plots of ground temperature (°C), derived from the boreholes JB-B-2 (a), JB-B-3 (b), JB-B-6 (c), and JB-B-9 (d) at the JB site along the China-Russia Crude Oil Pipeline (CRCOP) I in the northern part of Northeast China. The blank gap indicates the missing data.

### 3.2.3 Soil water content on the ROW of pipeline

The variations in VWC at depths of 0.5 m (peaty soil) and 1.5 m (silt clay) are controlled by the freeze-thaw

processes. In the ground thawing season, VWC is 55% at 0.5 m in depth and reaches 64% at 1.5 m. While the VWC
at 2.5 m (silt clay) is less variable with an average of 45% (Fig.10), offering indirect evidence to the presence of the
SST around the warm-oil pipe (Li et al., 2018). All these findings undoubtedly confirm that the construction and
operation of the buried warm pipeline have resulted in locally intensive thermal disturbances on the underlying
permafrost in the ROW along CRCOPs, although spatially confined.

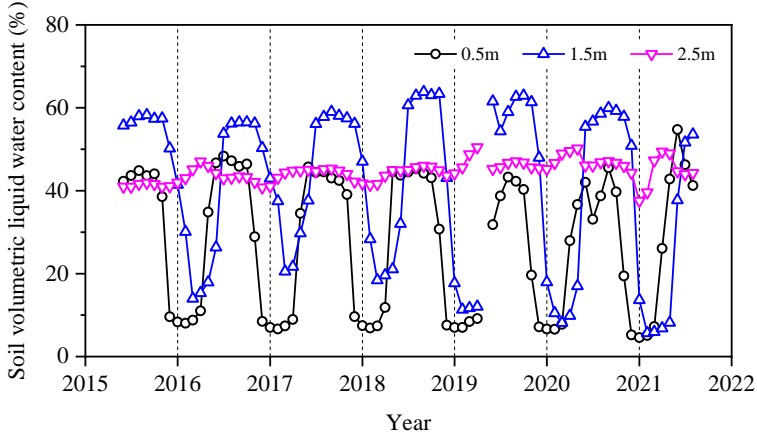


Figure 10. Temporal history of soil volumetric liquid water content at depths of 0.5, 1.5, and 2.5 m at the JB site along
the China-Russia Crude Oil Pipeline (CRCOP) I in the northern part of Northeast China during 2015-2021.

### 3.3 Subsurface electrical resistivity data

The electrical resistivity (ER) is dependent on many parameters including lithology, soil water/ice content, and soil
temperature. The ER distribution within the subsurface can be visualized by ERT. The inverted ERT results provide
a continuous transect of the characteristics of the active layer and near-surface permafrost to delineate the shape
and size of talik (unfrozen ground in permafrost regions) or permafrost islands along the CRCOPs route (Zhang,
2011). We performed ERT surveys in April 2018 with the SuperSting R8 system (Advanced Geosciences, Inc.,
Table 2) using the Wenner-Schlumberger configuration due to its high signal-to-noise ratio (Dahlin and Zhou,
2014). In addition to the fourth ERT profile at the JB site (P-JB-4), other profiles were done using stainless steel
electrodes spaced by 2 m along the 120-m-long profile, reaching a maximum penetration depth of 24 m (Table 3).
The smoothness-constrained least-squares method was employed for ERT inversion.
The configurations of talik around the CRCOPs can be seen in Figure 11. Here, an ER value of 300 Ωm was

used as the critical value to identify the boundary between frozen and unfrozen zones combined with the profile

characteristics of resistivity, GT, water/ice content, lithology (obtained from borehole drilling, Fig.12), and other

ERT surveys in Northeast China made by previous scholars (Hu and Shan, 2016; Li et al., 2021). There was a

significant difference in the sizes of taliks around the CRCOPs in April 2018, suggesting that pipeline operation-

related thermal disturbances had accelerated permafrost thaw (Fig. 11).

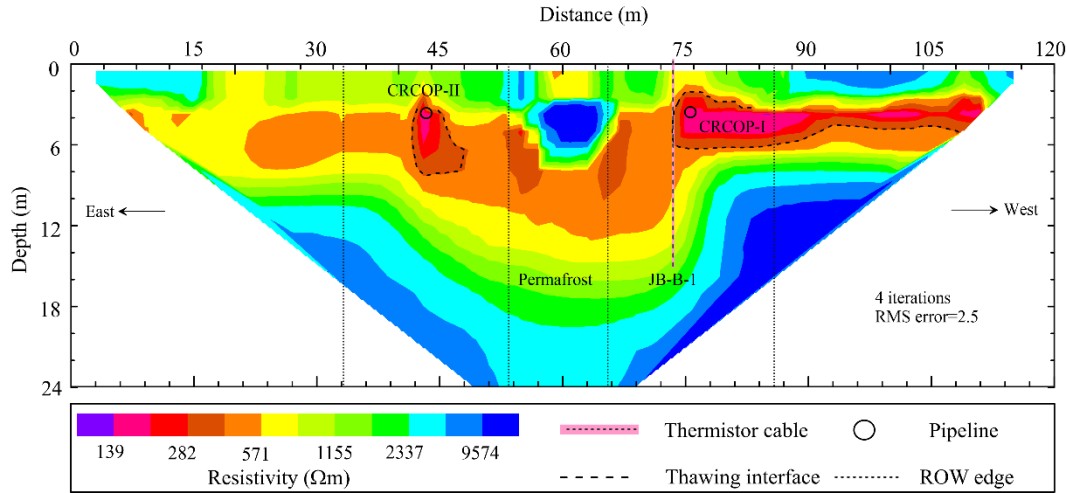

Figure 11. Inversion results of electrical imaging along P-JB-1 profile at the JB site along the China-Russia Crude Oil
Pipeline (CRCOP) I in the northern part of Northeast China in April 2018.

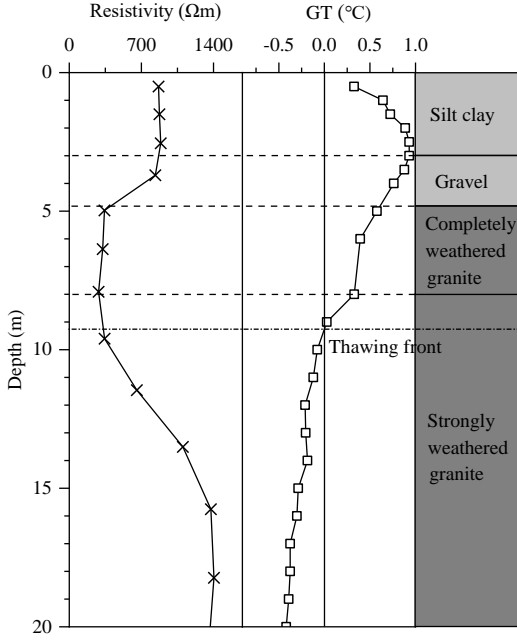

Figure 12. Dependence of electrical resistivity on ground temperature and lithology (in the borehole JB-B-1).

## 4 Data availability

The data sets presented herein can be freely downloaded from the National Tibetan Plateau/Third Pole Environment

Data Center at http://doi.org/10.11888/Cryos.tpdc.272357 (Li, 2022).

## 5 Conclusions

For this study, an *in-situ* monitoring network for the ground thermal state of permafrost was established along the

CRCOP route, at the eastern flank of the northern Da Xing'anling Mountains in Northeast China. The resulting

dataset fills the gaps in the spatial coverage of mid-latitude mountain permafrost databases with and without warm

pipeline disturbances. This dataset consists of daily ground temperatures at depths of 0-20 m in 20 boreholes (10.0

to 60.6 m deep, spanning a range of latitudes from 50.47 to 53.33°N), soil volumetric liquid water contents,

meteorological variables, and ERT data. The harsh environmental conditions and inaccessibility to the field sites

during the COVID-19 pandemic have resulted in some missing data, but we will continue to update the dataset by

overcoming these difficulties in subsequent years.

Analysis of data compiled indicates permafrost conditions along the eastern flank of the northern Da

Xing'anling Mountains are controlled by latitude and strongly influenced by the local geo-environmental factors.

The MAGT at 15 m depth ranges from −1.8 to −0.4 °C, and the ALT varies from 100 to 270 cm from north to south

in permafrost terrains. The record from 2011 to 2020 of GT measurements off the ROW indicates an extensive

ground warming in the vicinity of the southern limit of latitudinal permafrost. Permafrost temperatures at depths of

8–20 m have been rising at a rate of 0.035 °C yr$^{-1}$, but there is no significant change in ALT, varying between 178

and 200 cm in the 10-year observation period. The GT measurements on the ROW of the pipeline and the ERT

results show that the construction and operation of the CRCOP have brought strong thermal disturbances to the

underlying and ambient permafrost foundation soils, leading to a talik formation to a maximum depth of 11.5 m

around the pipeline, although laterally much confined to the ROW of the CRCOP I. This permafrost disturbance is

still expanding and can persist for decades. The permafrost beneath the pipeline ROW cannot be prevented but can
be significantly reduced by installing insulation or thermosyphons. This dataset provides a solid basis for assessing
the spatiotemporal variability of ground hydrothermal states of the active layer and near-surface (generally $\leq 20$ m)
permafrost under the linear disturbances of the buried warm pipeline and related environmental effects, for
revealing hydro-thermal-mechanical interactions between buried pipeline and the ambient permafrost environment,
for evaluating the integrity of the pipeline systems, and for offering crucial and badly needed data on the design,
construction and maintenance of similar pipelines in permafrost regions.

## Author contribution

GY, WM and HJ designed this study and obtained the financial support for establishing and maintaining the
monitoring sites. FW and GY compiled the dataset, performed the analysis, and wrote the manuscript. FA, YH, and
DC improved the writing. GW, YP, YZ, YC, JZ, KG, RH, XJ, XL, and YL participated in the fieldwork and editing
of manuscripts at various stages.

## Competing interests

The authors declared no conflicts of interest in this work.

## Special issue statement

This article is part of the special issue "*Extreme Environment Datasets for the Three Poles*". It is not associated
with a conference.

## Acknowledgments

This work was financially supported by the Strategic Priority Research Program of Chinese Academy of Sciences
(Grant No. XDA2003020102), the National Natural Science Foundation of China (Grant No. 42101121), and the
Research Projects of the State Key Laboratory of Frozen Soil Engineering (Grant Nos. SKLFSE-ZY-20 and
SKLFSE202010).

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

## Appendix A: Abbreviations

| | |
|---|---|
| ALT | Active layer thickness |
| CRCOP | China-Russia crude oil pipeline |
| ER | Electrical resistivity |
| ERT | Electrical resistivity tomography |
| GT | Ground temperature |
| MAAT | Mean annual air temperature |
| MAGT | Mean annual ground temperature |
| ROW | Right-of-way |
| SST | Supra-permafrost subaerial talik |
| VWC | Volumetric liquid water content |
| ZAA | Zero annual amplitude |

## Appendix B: Photos of meteorological station and instrumentations

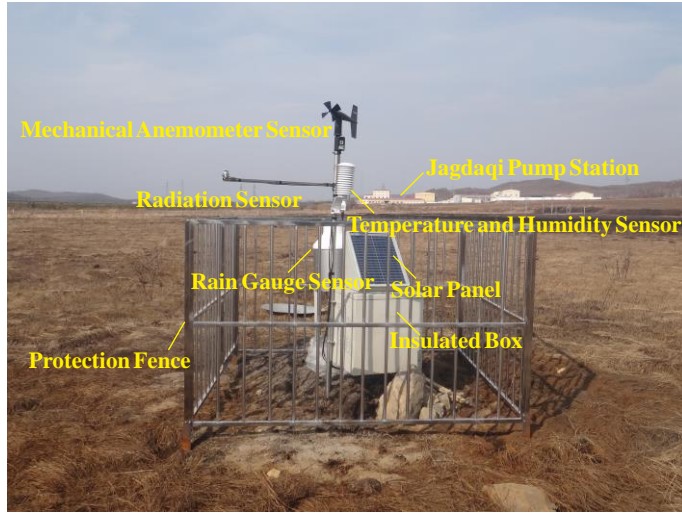

Figure B1. The automated weather station and instrumentations at the JB site along the China-Russia Crude Oil Pipelines route in the northern Da Xing'anling Mountains, Northeast China. Notes: Photo was taken on April 2018. The rain gauge sensor has been installed but is ineffective. The photo shows the location of the Jagdaqi pump station. The CR3000 data logger, multiplexer, battery cell, solar charge controller, and wireless transmission module are placed in the white box with a solar panel (*i.e.,* insulated box). All monitoring devices are protected by an aluminum alloy fence.

**Appendix C: Relative position of boreholes and automatic collection instrumentation for ground temperatures at permafrost monitoring sites**

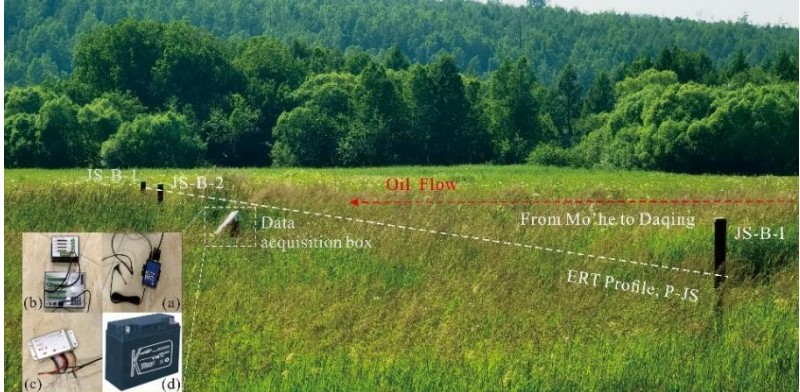

Figure C1. Position of boreholes drilled on and off the pipeline ROW and the ERT profile at the JS permafrost site. Photo was taken on 29 June 2021. Ground temperatures are measured using thermistor chains connected to the CR3000 data logger. Notes: (a) Wireless transmission module (HKT-DTU, Campbell Scientific, Inc., USA), (b) CR3000 data logger with a TRM128 multiplexer, (c) Solar charge controller (Phocos ECO (10 A), Germany), (d) Battery cell, a part of the power supply device.

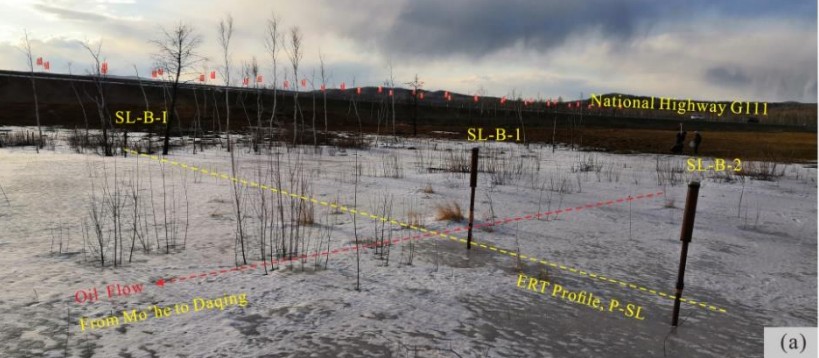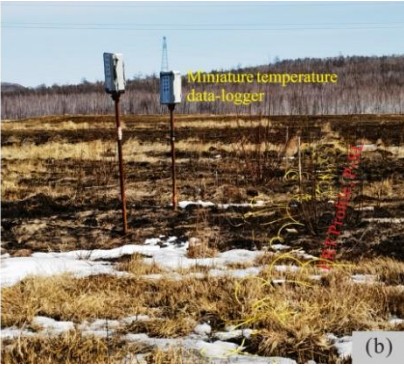

Figure C2. Position of boreholes drilled on and off the pipeline ROW and the ERT profile at the SL permafrost site. Notes: (a). Photo was taken on 30 March 2018. The ground surface within the trench is completely covered with ice and snow. (b) Miniature temperature data loggers were installed in August 2020. Photo was taken on 17 April 2021. The surface vegetation was destroyed by a controlled burn.

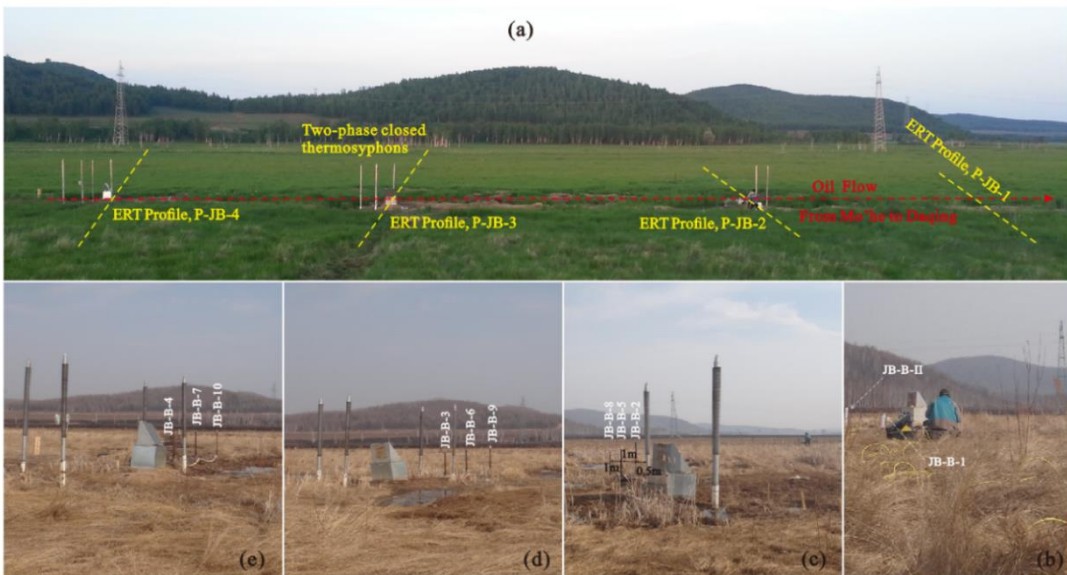

Figure C3. Picture of the monitored cross-sections, perpendicular to the pipeline at 20 m intervals, at the JB permafrost
site. Notes: (a) Plane view, (b) Section 1, without thermosyphon, (c) Section 2, one pair of thermosyphons, (d) Section 3,
two pairs of thermosyphons at a longitudinal spacing of 1.3 m, and (e) Section 4, two pairs of thermosyphons at a
longitudinal spacing of 1.4 m. The data acquisition device is the same as that at the JS site.

**Appendix D: ERT results along P-JS, P-SL, P-JB-2, P-JB-3, and P-JB-4 profiles**

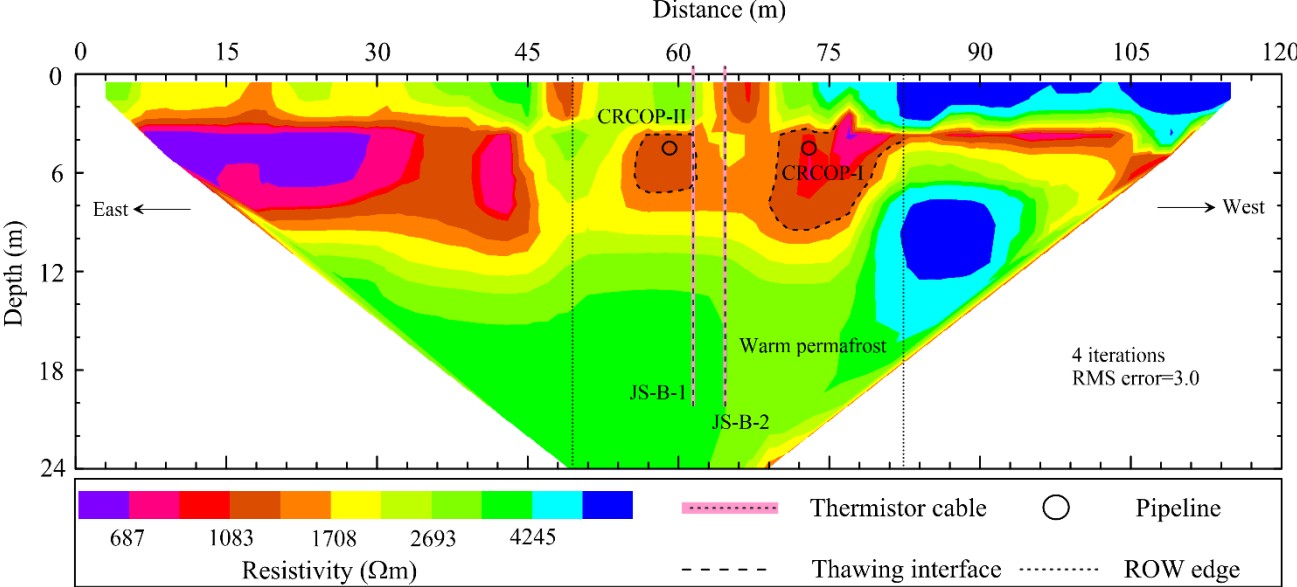

Figure D1. Inversion results of electrical imaging along P-JS profile at the JS site, carried out in April 2018.

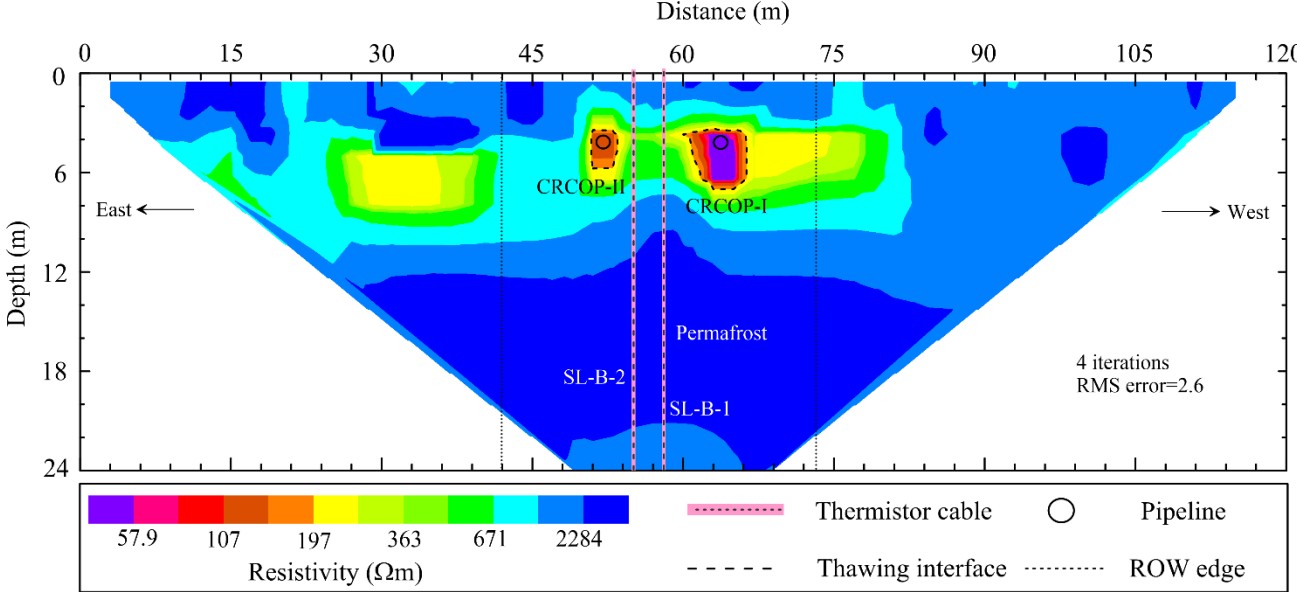

Figure D2. Inversion results of electrical imaging along P-SL profile at the SL site, carried out in April 2018.

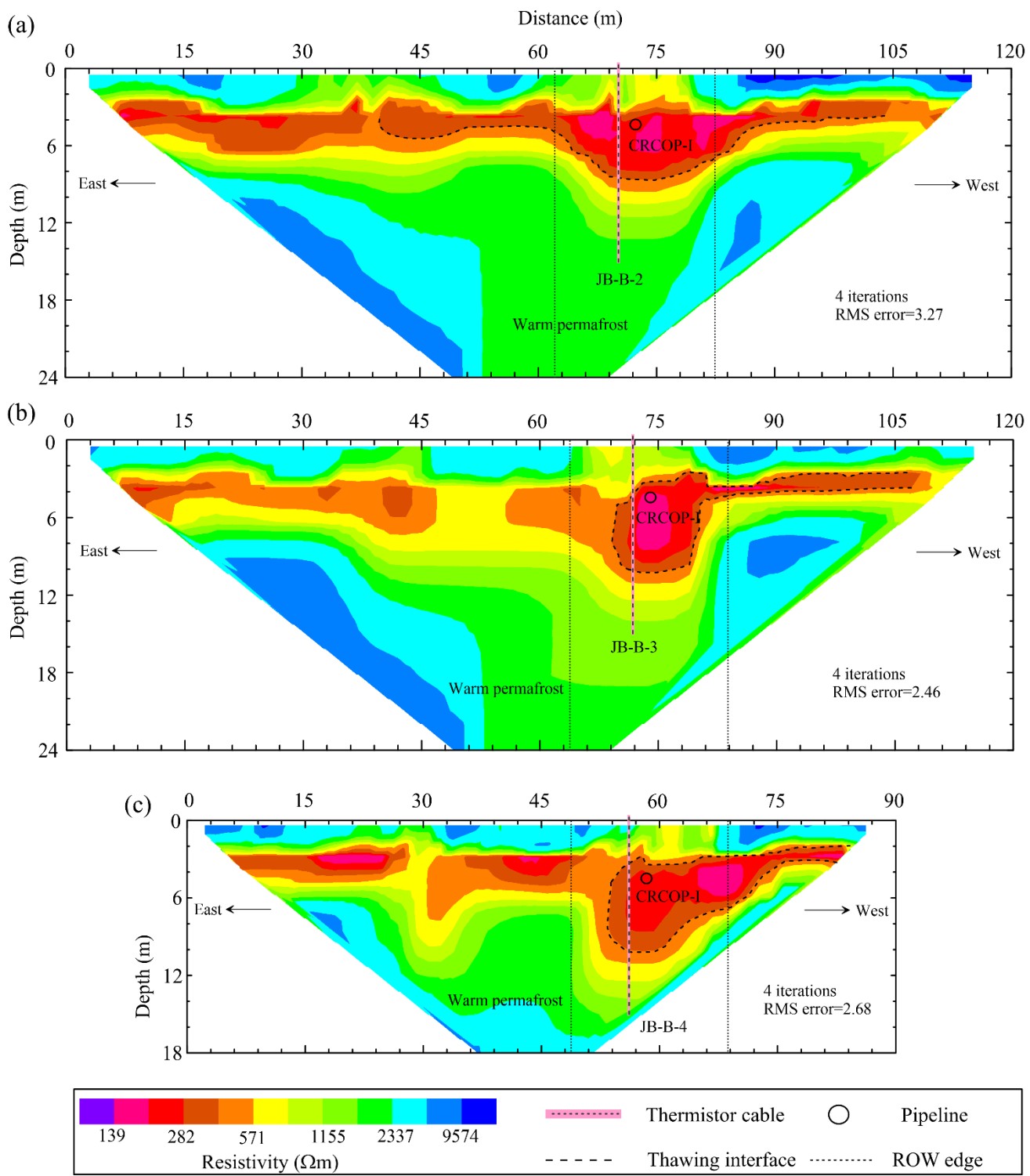

Figure D3. Inversion results of electrical imaging along monitored cross-sections with thermosyphons at the JB site, carried out in April 2018. Notes: (a) P-JB-2 profile, one pair of thermosyphons, (b) P-JB-3 profile, two pairs of thermosyphons at a longitudinal spacing of 1.3 m, and (c) P-JB-4 profile, two pairs of thermosyphons at a longitudinal spacing of 1.4 m.