# Peer review of "A newly integrated ground temperature dataset of permafrost along the China-Russia Crude Oil Pipeline route in Northeast China"

_Earth System Science Data, 2022_

## Author Comment (AC1)

**General Comments**

The paper submitted by Li et al. describes a permafrost dataset compiled from measurements made along an oil pipeline route in NE China. The dataset includes information on ground temperatures, soil moisture along with results of geophysical surveys and meteorological data. Instrumented sites are located at various distances from the pipeline and are both on the right-of-way (ROW) and in undisturbed terrain off the ROW which facilitates characterization of impacts of pipeline characterization, similar to that done along other pipelines (e.g. Burgess and Smith 2003; Burgess et al. 2010; Johnson and Hegdel 2008). Such datasets and associated analysis are valuable because they can be utilized for model calibration and validation and also to evaluate mitigation techniques to reduce impacts and improve design of future infrastructure projects (e.g. Burgess et al. 2010). For these reasons, the paper and associated database should of interest to engineers and those involved in environmental impact assessment. However the manuscript does require a number of revisions before it can be accepted for publication.

Response:
Thank you very much for your positive evaluation of our work. We have made a thorough revision to the original manuscript based on your constructive comments and suggestions, and here we give the responses one by one.

As mentioned above, this is not the first time monitoring programs have been established along pipeline routes in permafrost regions and the data utilized to evaluation design performance, effectiveness of mitigation techniques as well as the lessons learned being used for design of other projects. For example, there are several papers and reports on two pipeline corridors in NW North America (Norman Wells to Zama (NWZ) oil pipeline NWT Canada and Alyeska oil pipeline in Alaska). These include for example Burgess et al. (2010); Burgess and Smith (2003); Smith and Burgess (2010); Croft et al. (2021); Mosely et al. (2021); Johnson and Hegdel (2008), as well as other reports referenced in these. Data compilations are available for the NWZ pipeline (Smith et al. 2004, 2008) and may also be available for the Alyeska oil pipeline. It would be useful to refer to these other studies in both the introduction to provide the rationale for compilation of similar databases and also when interpreting the results of data analysis.

Response:
What you said is absolutely right. Pipelines constructed in permafrost have been inevitably faced to many problems related to thaw settlement, frost heave, slope failure, icing, and frost mounds, which were forced to establish monitoring systems for permafrost environmental and stability evaluation such as along the Norman Wells pipeline and the Trans-Alaska oil pipeline, and to obtain field data for understanding how the permafrost foundation performed when pipelines went through or above it. Correspondingly. Such datasets and associated analyses are valuable for this manuscript because they do provide references and implications for CRCOPs when preparing this manuscript. According to your suggestion, most of these studies have been introduced and referred to in this revised manuscript as follows:

As a result, a permafrost monitoring network along the CRCOPs route was gradually established

by referring to the experiences and lessons from other oil/gas pipelines (e.g. Norman Wells to Zama oil pipeline in Canada, Alyeska oil pipeline in the U.S., and Nadym–Pur–Taz natural gas pipeline in Russia) in permafrost regions (Burgess and Smith, 2003; Johnson and Hegdal, 2008; Smith and Riseborough, 2010; Oswell, 2011). Boreholes were instrumented to measure GTs in the active layer and near-surface permafrost on and off the right-of-way (ROW) of the CRCOPs and electrical resistivity tomography (ERT) surveys were used to delineate frozen and unfrozen ground in the vicinity of the CRCOPs (Kneisel et al., 2008; Farzamian et al., 2020).

Newly added references as follows:
*Burgess M M, Smith S L. 17 years of thaw penetration and surface settlement observations in permafrost terrain along the Norman Wells pipeline, Northwest Territories, Canada. Proceedings of the Eighth International Conference on Permafrost, 2003: 107-112.*
*Johnson E R, Hegdal L A. Permafrost-related performance of the Trans-Alaska oil pipeline[C]. Proceedings of 9th International Conference on Permafrost, Fairbanks, AK, USA. 2008: 857-864.*
*Smith S L, Riseborough D W. Modelling the thermal response of permafrost terrain to right-of-way disturbance and climate warming[J]. Cold Regions Science and Technology, 2010, 60(1): 92-103.*
*Oswell J M. Pipelines in permafrost: geotechnical issues and lessons[J]. Canadian Geotechnical Journal, 2011, 48(9): 1412-1431.*

The paper could be better organized and some important information should be provided. In particular, Section 2 (Study site and instrumentation) could be better written and include some information from Section 3 (Data description). It would be better to clearly describe the instrumentation used to measure each variable (i.e. ground temperature, moisture content, air temperature etc.) and use a table to indicate which variables are measured at each site/borehole. This would reduce repetition and enhance clarity. Some of the information in Section 3 also describes instrumentation and should be moved to Section 2 so that all methods appear in the same section.

Response:
Thank you very much for your comments. In the revised manuscript, Section 2 (Site description) and Section 3 (Data description) have been re-organized referring to your comments and the recently published papers in ESSD (e.g. Boike et al., 2019; Wu et al., 2022). Section 3 is divided into three sub-sections: meteorological data, ground temperature and soil water content data, and subsurface electrical resistivity data.

Information on distance of boreholes from the pipe centre line for those on the ROW needs to be clearly provided. Also, the ROW width at each site should be provided as well as the distance from the ROW edge for boreholes that are in the undisturbed terrain off the ROW. This information is important as it provides information on the amount of disturbance there might be at each site. This information could be provided in a table or as site plans in the supplementary information.

Response:
Thanks for your better suggestions. We are sorry that we did not provide the exact locations of

boreholes and the pipeline ROW width. The ROW width of the CRCOPs is approximately 20 m along its route. The distances of boreholes from the pipe centerline of CRCOP I have been added in Table 3 in the revised manuscript as follows. Four boreholes of JS-B-1, JS-B-2, SL-B-1, and SL-B-2 were drilled on the ROW of CRCOP II. Their distances from the centerline of CRCOP II are provided. In turn, the distances of off-ROW boreholes from the ROW edge are easily determined.

Table 3 Summary of monitoring information of ground temperature boreholes, water content monitoring pits, and ERT profiles along the CRCOPs in Northeast China.

| Variable | Borehole/ ERT profile | Maximum monitoring depth (m) | Distance from pipe centreline (m) | Data logger | Measuring internal | Operation period |
|---|---|---|---|---|---|---|
| Soil/permafrost temperature at the natural site (off-ROW) | XA-B-I | 60.5 | 100 | CR 3000 | 2h, AUTO | Nov 2018 – Nov 2020 |
| | XT-B-I | 20 | 80 | RTB37a36V3 | 2h, AUTO | Jul 2019 – Aug 2021 |
| | JS-B-I | 20 | 24.8 | CR 3000 | 2h, AUTO | Dec 2018 – Jun 2021 |
| | SL-B-I | 25 | 12.6 | Fluke 87/89 RTB37a36V3 | Monthly, MANU 2h, AUTO | Sep 2017 – Oct 2019 Aug 2020 – Dec 2020 |
| | JB-B-I | 60.6 | 80 | CR 3000 | 2h, AUTO | Jun 2018 – Aug 2020 |
| | JB-B-II | 20 | 16.6 | Fluke 87/89 CR 3000 | Monthly, MANU 2h, AUTO | Nov 2011 – Sep 2017 Oct 2017 – Aug 2021 |
| Soil/permafrost temperature on pipeline ROW (on-ROW) | JS-B-1* | 19.8 | 2 | CR 3000 | 2h, AUTO | Oct 2017 – May 2021 |
| | JS-B-2* | 20 | 5 | CR 3000 | 2h, AUTO | Oct 2017 – Aug 2021 |
| | SL-B-1* | 24.8 | 3 | Fluke 87/89 RTB37a36V3 | Monthly, MANU 2h, AUTO | Sep 2017 – Oct 2019 Aug 2020 – May 2021 |
| | SL-B-2* | 24.8 | 5.9 | Fluke 87/89 RTB37a36V3 | Monthly, MANU 2h, AUTO | Sep 2017 – Oct 2019 Aug 2020 – May 2021 |
| | JB-B-1 | 20 | 2 | Fluke 87/89 CR 3000 | Monthly, MANU 2h, AUTO | Mar 2014 – Sep 2017 Oct 2017 – Aug 2021 |
| | JB-B-2 | 15 | 2 | CR 3000 | 2h, AUTO | Jun 2015 – Aug 2021 |
| | JB-B-3 | 15 | 2 | | 2h, AUTO | Jun 2015 – May 2018 |
| | JB-B-4 | 15 | 2 | | 2h, AUTO | Jun 2015 – May 2019 |
| | JB-B-5 | 10 | 3 | | 2h, AUTO | Jun 2015 – Aug 2021 |
| | JB-B-6 | 14 | 3 | | 2h, AUTO | Jun 2015 – May 2018 |
| | JB-B-7 | 15 | 3 | | 2h, AUTO | Jun 2015 – May 2020 |
| | JB-B-8 | 15 | 4 | | 2h, AUTO | Jun 2015 – Aug 2021 |
| | JB-B-9 | 15 | 4 | | 2h, AUTO | Jun 2015 – May 2018 |
| | JB-B-10 | 15 | 4 | | 2h, AUTO | Jun 2015 – May 2020 |
| Soil volumetric liquid water content on pipeline ROW | JB-W1 JB-W2 JB-W3 | 2.5 | 1 | CR 3000 | 2h, AUTO | Jun 2015–Aug 2021 |
| Electrical resistivity | P-JS | 24 | | SuperSting R8 system | Site visit | Apr 11, 2018 |
| | P-SL | 24 | | | Site visit | Apr 12, 2018 |
| | P-JB-1 | 24 | | | Site visit | Apr 06 – Apr 10, 2018 |
| | P-JB-2 | 24 | | | | |
| | P-JB-3 | 24 | | | | |
| | P-JB-4 | 18 | | | | |

* Boreholes were drilled on the ROW of CRCOP II. The ROW width along the pipeline is about 20 m.

Section 3 doesn't really provide a clear description of the data in the dataset but rather provides some information on instrumentation as well as presentation of some of the data in graphic form and interpretation of the results. It would be useful to have a brief and clear description of the data. For example, are hourly, daily, annual mean values provided; if annual values provided, what is the time period (calendar or hydrologic year) and how do you deal with missing values; describe any processing done to ensure data quality and any derived parameters that might be included in the dataset. The presentation of the data and interpretation of results could be in a separate section.

Response:
Section 3 has been revised to improve the description of the data according to your comments. For example, we have added the description of quality control of data in the revised manuscript. Obvious erroneous recordings are manually removed and all the missing or abnormal data are replaced with null values before daily average values are calculated. All meteorological data from the small automated AWS, GT data at different depths in boreholes, and VWC data, at the daily resolution, have been complied in this dataset.

A number of specific comments are provided below. Some of these are editorial while others are related to clarity of the content. I hope the authors find the comments helpful in preparing an improved manuscript that should be acceptable for publication.

**Specific Comments**

L19-20 – suggested revision: "In 2011 we initiated a…."

Response:
It was revised according to your suggestion.

L21-24 – Not clear. Do you mean you compiled an integrated dataset on the ground thermal state utilizing the various types of data that are mentioned in the first part of the sentence. You should revise to be clear.

Response:
The sentence has been revised as "We compiled an integrated dataset of the ground thermal state along the CRCOPs route, consisting of meteorological data near the southern limit of latitudinal permafrost (SLLP), ground temperature data in 20 boreholes with depths of 10–60.6 m, soil volumetric liquid water contents and 2-D electrical resistivity tomography (ERT) data at different sites.".

L24–26 – Are you referring to the undisturbed sites off the ROW? Revise to be clear. A revision is also suggested: "Results demonstrate that permafrost has warmed between 2011-2020 in the vicinity of SLLP, manifested as an increase in ground temperature at depth likely in response to climate change."

Response:

Thanks. The sentence has been revised according to your suggestion.

L28 – Insert "an: between "as" and "insulation"

Response:
Added.

L37-38 – I suggest you remove the older references (ones between 1999 and 2008) and add more recent ones such as Etzelmuller et al. (2020); Zhao et al. (2020) and Liu et al. (2021). You could also cite the most recent BAMS State of the Climate sections on permafrost (i.e. Smith et al. 2021 and Noetzli et al. 2021).
L39 – You could delete the older Zhang et al. ref and add newer ones such as Burke et al. (2020) and other recent papers (you could also cite most recent IPCC reports of 6th Assessment).
L41 – I think you can delete the older references here (before 2010) as you have several recent ones.

Response:
According to your suggestions, the older references have been deleted and more recent references have been added. The sentence has been revised as "Over the last few decades, the warming and thawing of permafrost have been observed in most permafrost regions (e.g., Ran et al., 2018; Biskaborn et al., 2019; O'Neill et al., 2019; Etzelmüller et al., 2020; Liu et al., 2021; Noetzli et al., 2021; Smith et al., 2022), and permafrost degradation will continue in response to a warming climate (Koven et al., 2013; Burke et al., 2020). Permafrost change affects the geomorphological characteristics, carbon release, hydrological process, ecosystem, climate system, and integrity of infrastructure (Cheng and Jin, 2013; Beck et al., 2015; Hjort et al., 2018, 2022; Turetsky et al., 2020; Jin and Ma, 2021; Jin et al., 2021, 2022; Luo et al., 2021; Jones et al., 2022; Liu et al., 2022; Miner et al., 2022).".

L70-71 – Permafrost presence/absence is inferred or interpreted from the ERT results rather than "detected". A revision is suggested: "… electrical resistivity tomography (ERT) surveys were used to determine the distribution of frozen and unfrozen ground in the vicinity of the CCOPs……ERT surveys to characterize freeze-thaw dynamics in…."

Response:
We are sorry for this unreasonable description. The sentence has been revised according to the suggestion.

L68-74 – I don't think you need this much detail here as it should be covered in the methods section (section 2). You could just say that boreholes were instrumented to measure ground temperatures and ERT surveys were used to delineate frozen and unfrozen ground.

Response:
According to your suggestion, the sentence has been revised as "Boreholes were instrumented to measure GTs in the active layer and near-surface permafrost on and off the right-of-way (ROW) of

the CRCOPs and electrical resistivity tomography (ERT) surveys were used to delineate frozen and unfrozen ground in the vicinity of the CRCOPs (Kneisel et al., 2008; Farzamian et al., 2020).".

L82 – replace "are" with "were"

Response:
It was revised.

L88-90 – Revision suggested "Between 1972 and 2017, MAAT increased at a …" You can also delete "climate wetting".

Response:
According to the suggestion, the sentence has been revised as "Between 1972 and 2017, MAAT increased at a rate of 0.32 °C per decade while annual precipitation increased at a rate of 14.6 mm per decade.".

L91 – You can delete the last part of the sentence from "retrieved" onward. The additional information can be provided in the Table (or its caption) if you need to provide references to previous studies.

Response:
It was deleted according to your comment .

L92-93 – Suggested revision – "Permafrost is warm with mean annual ground temperature…….ranging from -1.8 to -0.4°C. The reference may not be necessary as it seems to be a textbook, rather than a reference that specifically reports on MAGT in the region – perhaps one of the other papers in the list would be more suitable.

Response:
The sentence has been revised according to the suggestion and the less related reference was deleted.

L95-97 – Suggested revision – "The XA site, located in a permafrost wetland, is the most northern site and has the lowest air temperature, while……China has the highest air temperature and permafrost occurs in isolated patches.

Response:
The sentence has been revised according to the suggestion.

L99-100 – Can't you just say it is ice-rich permafrost?

Response:
It was revised as "ice-rich permafrost".

L108 – replace "built" with "installed"

Response:
It was revised.

L111 – You could just say when the hole was drilled and instrumentation installed. It isn't clear why it is important to say that it was finished 6 months before installation of AWS. If it was a long drilling time and likely caused significant disturbance to the thermal regime just give the number of days of drilling and the implication of this.

Response:
We are sorry for this vague statement. The sentence has been revised as "Besides, a new borehole (JB-B-I) was drilled in March 2017 down to 60.6 m near the above-mentioned AWS.".

L119 – How far off the ROW were these sites – this information should be provided.

Response:
The distances of off-ROW boreholes at each site from the centerline of CRCOP I were added in Table 3 in the revised manuscript.

L118-139 – Some of the detail could be reduced by putting some of the information in a table (or figure) such as the distance from the pipe, or depth of borehole. The table could also summarize all the instrumentation or surveys at a site, i.e. add this information to Table 3 for example. In this section you could just mention the instrumentation used to acquire information on the various parameters such as GT, soil moisture etc. You should also indicate how wide the ROW is at each site as this will give the reader an indication of the amount of the disturbance.

Response:
This paragraph has been rewritten and some necessary information has been added in Table 3. Details can be found in the above replies to the comment.

L122-123 – Don't you mean the boreholes were drilled vertically? (rearrange sentence)

Response:
Sorry for the unclear statement. Yes, all boreholes were drilled vertically. This sentence has been revised as "At the SL site, two on-ROW boreholes (SL-B-1 and SL-B-2, 3 and 5.9 m from the centerline of CRCOP II, respectively) and one off-ROW borehole (SL-B-I, 2.6 m from the edge of the CRCOP I ROW) were drilled in March 2017 and instrumented in September 2017.".

L129 – Weren't thermistor cables used in other BHs?

Response:
The GT measurement was carried out by installing a thermistor cable protected by a steel tube into

the borehole. When a borehole was completed, a steel tube was installed in it. Then, the borehole outside the steel tube was densely backfilled with fine-grained sand and a thermistor cable was placed in the steel tube. Thermistor cables containing thermistors at designed intervals were used to monitor ground temperatures.

L134 – Do you mean precision (resolution) rather accuracy here? Wouldn't accuracy be more related to the temperature sensor?

Response:
Agree. In the revised version, we changed this sentence to "…generally has a lower resolution than the CR3000 data logger (±0.05°C).".

L148 Table 2 – The resolution/precision of instrumentation, including the dataloggers used should also be provided.

Response:
The accuracy of the manual readings is estimated to be ±0.1 °C, which is lower than the CR3000 data logger (±0.05°C) (Juliussen et al., 2010). The GT data collected by the RTB37a36V3 data logger is not as steady as that collected by the CR3000 data logger, i.e., the RTB37a36V3 data logger has a lower resolution than the CR3000 data logger. This content has been added in the revised manuscript.

L151 – Table 3 – You could just give the maximum measurement depth in the table as the various measurement depths would be included in the database. Alternatively, you could include this additional information as a supplementary information table. You could then include the additional information in the table suggested in an earlier comment (distance from pipe, ROW width at each site, and whether BH is on or off ROW).

Response:
Table 3 has been improved and revised according to your comments.

L153 – Section 3 - Data description – You seem to be confusing methodology with dataset description. Much of this information including type of instrumentation used should be included in Section 2 which describes methods including instrumentation. In some cases such as Section 3.2 there is a repeat of the instrumentation description that was given in Section 2. Section 3 also presents results, analysis and discussion so some consideration should be given to revising the organization of sections.

Response:
According to your suggestions, the paper was re-organized and section 3 has been revised to improve the description of the data. Details can be found in the above responses.

L190 – Fig. 3b doesn't clearly show the relationship between GT and latitude and whether it is linear or not. MAGT at DZAA for each undisturbed site could be plotted vs latitude to better show

this.

Response:

As shown in Figure 3b, the annual changes of ground temperatures at the depth of 15m at all monitoring sites are negligible except for the JS site. This depth is considered as the depth of zero annual amplitude (DZZA). The MAGTs at DZAA at the sites of JB, SL, XT, and XA are −0.7, −0.8, −1.8, and −1.7°C respectively, indicating that the MAGT at DZAA is greatly affected by the latitude, but the relationship between MAGT and latitude is not linearly dependent.

[Figure]

L192-193 – Are you sure this isn't an issue with the sensors?

Response:
We think it is not the failure of sensors. Firstly, the thermistor cable used to measure GT has a high accuracy of ± 0.05 °C and the measuring range of the thermistor ranges from −30 to 30 °C. The collected data thus are reliable. Secondly, this abnormal phenomenon has occurred at depths of 10, 15, and 20 m for two consecutive years, which would seem to indicate that the failure of the thermistors is not possible. In addition, the strata of this site (obtained from the JS-B-I borehole) are composed of peaty soil (0-0.8 m), grey silty clay (0.8-2 m), yellowish-brown gravel (2-5.5 m), and strong weathered granite (5.5-20 m). The presence of the rock layers with high permeability provides a channel for the movement of intra-permafrost water.

L197-199 – A bit confusing as Fig. 4 shows seasonal variation.

Response:
It was changed in the revised manuscript.

L199-200 – Are you referring to the thermal offset here?

Response:
We describe the characteristics of the ground temperature curve. As for the thermal offset, it is not discussed. At depths from 1 to 2 m, the monthly average GTs in 2018 was fluctuating in proximity to 0 °C.

L201-204 – A couple of other things to consider: ALT also responds more to shorter-term

variations in air temperature compared to the deeper ground temperatures for which the higher frequency variations are filtered out. ALT will therefore show more interannual variability as shown in Fig. 5. The ground at JB is ice-rich, and melting of ground ice as thaw progresses deeper into the ground can lead to surface subsidence and consolidation of the unfrozen material. Changes in ALT over time can be relatively small as ice-rich material thaws compared to sites where excess ice is negligible. See for example Nyland et al. (2021), O'Neill et al. (2019).

L217-218 – The key thing is that insulation doesn't prevent heat transfer but reduces it.

Response:
Thanks for your comments. The above questions have been considered carefully and interpreted in revising the manuscript. Meanwhile, the relevant references are added.

L219-221 – As well as the effect of the pipe, any ROW disturbance such as changes to the surface, clearing of vegetation will also have an effect on the ground thermal regime. These changes could also effect the ground thermal regime off ROW (lateral heat transfer). See for example some of the publications on the Norman Wells pipeline mentioned earlier and also Burgess and Smith (2003), Smith and Riseborough (2010).

Response:
What you said is right. Our results also show that many factors lead to permafrost thawing and thaw settlement along the pipeline route including ROW vegetation clearing, trenching, warm oil temperature, surface water infiltration, groundwater flowing, and climate warming. Besides, the thermal effects of ROW disturbance can extend to the adjacent undisturbed terrain.

L225 – It isn't clear why this is an artificial permafrost table – is this the site with thermosyphons? If so, it should be mentioned first.

Response:
We are sorry for this mistake. It was changed to permafrost table.

L228-230 – Are you referring to surface settlement (subsidence) here?

Response:
Here it refers to the total settlement of pipeline foundation soil.

L245 – Figure 8 – Consider reversing the colour scale and using blue for higher resistivity (colder ground –permafrost) and red for lower resistivity (warmer ground – unfrozen). This is more intuitive and has been done in other papers presenting ERT results. If some of the transect is off ROW, indicate the ROW edge. Was there a topographic survey done? – no change in ground elevation is shown.

Response:
Thank you for your comments. We re-draw the figure and appendix D based on your suggestion.

The ERT surveys have been conducted at JB, SL, and JS sites. These three sites are selected on relatively flat terrain and there are no obvious changes in ground elevation, which is also described in the revised manuscript.

L246 – There acronym TPCT isn't really needed – just say thermosyphons in the rest of the paragraph.

Response:
It has been changed.

L246-262 – The papers mentioned earlier on the Alaska Alyeska oil pipeline might be relevant here as the effect of thermosyphons are described.

Response:
Yes, the relevant references have been added.

L269 – Volumetric moisture content doesn't really have units. (dimensionless) You could give as %.

Response:
Yes, we added the unit in Figure 9 in the revised version.

**References**

Burgess MM, Oswell J, Smith SL 2010. Government-industry collaborative monitoring of a pipeline in permafrost – the Norman Wells Pipeline experience, Canada. In: GEO2010, 63rd Canadian Geotechnical Conference and the 6th Canadian Permafrost Conference, Calgary, Sept 2010. GEO2010 Calgary Organizing Committee, pp 579-586. https://www.aina.ucalgary.ca/scripts/mwimain.dll/116/2/3/72659?RECORD&DATABASE=CPC

Burgess MM, Smith SL 2003. 17 years of thaw penetration and surface settlement observations in permafrost terrain along the Norman Wells pipeline, Northwest Territories, Canada. In: Phillips M, Springman SM, Arenson LU (eds) Proceedings of 8th International Conference on Permafrost, Zurich Switzerland, July 2003. A.A. Balkema, pp 107-112

Burke EJ, Zhang Y, Krinner G (2020) Evaluating permafrost physics in the Coupled Model Intercomparison Project 6 (CMIP6) models and their sensitivity to climate change. The Cryosphere 14:3155-3174. doi:10.5194/tc-14-3155-2020

Croft PE et al. 2021. Slope Stabilization along a Buried Crude-Oil Pipeline in Ice-Rich Permafrost. Proceedings of The Regional Conference on Permafrost 2021 and the 19th International Conference on Cold Regions Engineering, American Society of Civil Engineers. p. 339-350 https://doi.org/10.1061/9780784483589

Etzelmüller, B. et al. (2020). Twenty years of European mountain permafrost dynamics — the PACE legacy. Environ. Res. Lett. 15, 104070.

Johnson ER and Hegdel LA 2008. Permafrost related performance of the Trans Alaska oil pipeline. Proc. 9th Int. Conf. on Permafrost, Fairbanks Alaska. 857-864

Liu et al. 2021. Permafrost warming near the northern limit of permafrost on the Qinghai–Tibetan Plateau during the period from 2005 to 2017: A case study in the Xidatan area. Permafrost and Periglacial Processes 32:323–334. https://doi.org/10.1002/ppp.2089

Mosley, L et al. 2021. Alyeska's 40-Plus Years of Experience with Heat Pipes on the Trans-Alaska Pipeline System. Proceedings of The Regional Conference on Permafrost 2021 and the 19th International Conference on Cold Regions Engineering, American Society of Civil Engineers. p. 327-338   https://doi.org/10.1061/9780784483589.031

Noetzli J, Christiansen HH, Hrbáĉek F, Isaksen K, Smith SL, Zhao L, Streletskiy DA (2021) [Global Climate] Permafrost Thermal State [in "State of the Climate in 2020"]. Bulletin of the American Meteorological Society 102 (8):S42-S44. doi:10.1175/BAMS-D-21-0098.1

Nyland, K.E., Shiklomanov, N.I., Streletskiy, D.A., Nelson, F.E., Klene, A.E., Kholodov, A.L., 2021. Long-term Circumpolar Active Layer Monitoring (CALM) program observations in Northern Alaskan tundra. Polar Geogr. 44, 167–185. https://doi.org/10.1080/1088937X.2021.1988000

O'Neill, H.B., Smith, S.L., Duchesne, C., 2019a. Long-term permafrost degradation and thermokarst subsidence in the Mackenzie Delta area indicated by thaw tube measurements, in: Cold Regions Engineering 2019. American Society of Civil Engineers, Reston, VA, pp. 643–651. doi:10.1061/9780784482599

Smith SL, Romanovsky VE, Isaksen K, Nyland KE, Kholodov AL, Shiklomanov NI, Streletskiy DA, Farquharson LM, Drozdov DS, Malkova GV, Christiansen HH (2021) [Arctic] Permafrost [in "State of the Climate in 2020"]. Bulletin of the American Meteorological Society 102 (8):S293-S297. doi:10.1175/BAMS-D-21-0086.1

Smith SL, Burgess MM 2010. Long-term field observations of cyclical and cumulative pipe and ground movements in permafrost terrain, Norman Wells Pipeline, Northwest Territories Canada. In: GEO2010, 63rd Canadian Geotechnical Conference and the 6th Canadian Permafrost Conference, Calgary, Sept. 2010. GEO2010 Calgary Organizing Committee, pp 595 https://www.aina.ucalgary.ca/scripts/mwimain.dll/116/2/4/72661?RECORD&DATABASE=CPC-602

Smith SL, Riseborough DW (2010) Modelling the thermal response of permafrost terrain to right-of-way disturbance and climate warming. Cold Regions Science and Technology 60 (1):92-103

Smith SL, Burgess MM, Riseborough D, Chartrand J (2008) Permafrost and terrain research and monitoring sites of the Norman Wells to Zama pipeline – Thermal data collection and case histories, April 1985 to September 2001. Geological Survey of Canada Open File 5331. doi:10.4095/224831

Smith SL, Burgess MM, Riseborough D, Coultish T, Chartrand J (2004) Digital Summary Database of Permafrost and Thermal Conditions – Norman Wells Pipeline Study Sites. Geological Survey of Canada Open File 4635. doi:10.4095/215482

Zhao, L. et al. (2020) Changing climate and the permafrost environment on the Qinghai–Tibet (Xizang) plateau. Permafrost and Periglac Process 31, 396–405.

---

## Author Comment (AC2)

**General comments**

This paper presents an integrated dataset on the permafrost along the China Russia crude oil pipeline in northeast China, which is an important complement to the current global permafrost database. These data are of great significance to the stability of linear infrastructures (e.g., pipeline, highway, railway), the validation of numerical simulation models, as well as the prediction of permafrost evolution process (degradation or aggradation). However, the manuscript still needs some revisions to reach a publishable level. My detailed comments as well as some suggestions are listed as below. Hope these comments will be useful to the authors to improve the manuscript.

Response:
Thank you very much for your comments. We have made a thorough revision to the original manuscript based on your comments and suggestions.

**Detailed comments**

Line 35: It should be better to say "the Earth's cryosphere".
Line 35-36, rewrite, "the thermal state" of permafrost" is not a component of cryosphere
Line 67: State Key Laboratory of Frozen Soil Engineering (SKLFSE), not soils. Check
Line 69: Delete the definite article "the" before electrical resistivity tomography (ERT).
Line 76: You can delete the comma before "in Northeast China", or keep the comma, but delete "in" before Northeast China.
Line 82: "…… were established ……"
Line 84: "…… was primarily based on ……"
Line 90: Delete "climate wetting at".
Line 108: "…… was installed ……"

Response:
Thanks for the suggestions, these sentences have been revised and highlighted in red in the revised manuscript.

Line 110-111: The sentence in the parenthesis is confusing. Do you mean the borehole was drilled 6 months before the installation of AWS? If so, you can just give the exact date when the bole was drilled to avoid any confusion.

Response:
We are sorry for this unclear statement. The sentence has been revised as "Besides, a new borehole (JB-B-I) was drilled in March 2017 down to 60.6 m near the above-mentioned AWS." according to your comments.

Line 116: Do you mean "wireless data transmission"? This also appears in line 157. Check!

Response:
It was changed as "Using such technology, it would be possible to check collected data in

real-time and identify possible sensor failures.".

Line 145: "…… other profiles were done using ……"
Line 147: "…… least-squares method was employed for ……"
Line 181: "…… protected by steel tube ……"
Line 187: Be careful to use "would be" herein, you can just say "…… which was mainly related to the local topography ……"
Line 193: Should say intra-permafrost groundwater.

Response:
Thanks for the comments, the above sentences have been revised and highlighted in red in the revised manuscript.

Figure 3. I noticed that there are "zero curtain" phenomena in the left panel (a1-a4) of Fig. 3, but the duration time of zero curtain of each curve is not the same. The authors should explain the possible reasons for this (I suppose this is mainly related to the in situ soil water/ice content of the permafrost sites).

Response:
The reason for different durations of zero curtain was added and highlighted in red in the revised manuscript.

Figure 6: To facilitate comparison, I suggest using the same range of values in the vertical axis (GT) for each subfigure. (Same for the left and right panels in Fig. 3)

Response:
We re-draw the Figure 6 according to your suggestion.

Line 228-229: This deepening of the permafrost table and warming of permafrost has exposed the pipelines to thawed low-bearing foundation soils. This sentence has two subjects, so you should use have instead of has.
Line 230: the CRCOP-I had locally settled down ……
Line 233: In the parenthesis, you can refer to the literature Zhang, T. (2011) when explaining "talik".
Zhang, T., 2011. Talik. In: Singh, V.P., Singh, P., Haritashya, U.K. (eds) Encyclopedia of Snow, Ice and Glaciers. Encyclopedia of Earth Sciences Series. Springer, Dordrecht. https://doi.org/10.1007/978-90-481-2642-2_563

Response:
The above sentences have been revised and relevant references have been added in the revised manuscript according to your comments.

Line 234-236: Why you used 300 ohm·m here? In my knowledge, this value can vary greatly from place to place around the world. The authors should give a criterion or any references to support

their choice (e.g., Schön, J. H., 1996; Christof Kneisel et al., 2008). Also, you said "boundary", it is a little bit confusing. I understand you mean the boundary between unfrozen and frozen zones. So, my suggested wording is: electrical resistivity value of 300 ohm·m was used as the critical value to identify the boundary between frozen and unfrozen zones.

Schön, J. H. (1996). Physical properties of rocks. Handbook of geophysical exploration. Pergamon Press.
Christof Kneisel, Christian Hauck, Richard Fortier, & Brian Moorman, 2008. Advances in Geophysical Methods for Permafrost Investigations, Permafrost and Periglacial Processes, 19: 157–178, DOI: 10.1002/ppp.616

Response:

What you said is right. We selected a threshold value of electrical resistivity for discriminating between frozen and unfrozen zones according to the relationship among the resistivity, ground temperature, and lithology, as shown below in Figure. The related description has been revised according to your comment.

[Figure]

Line 254: Lower rate, not slower rate.

Line 256: Delete "the" before "cooling rate".

Line 257-258: "abnormal changes in 0 °C isotherm observed in Figures 9b, 9c, and 9d ……" Here, the using of "observed" is strange to me. Figures can never observe anything by itself. So, just say "showed in figures 9b, 9c ……"

Line 259: intra-permafrost groundwater.

Figure 10: Should give the unit for VWC, m3/m3 or %.

Line 287: The MAGT at 15 m depth……

Line 289: I suggest using "occurred" rather than "observed".

Line 292: the ERT results

Response:

We have checked and revised the above English writing problems in manuscript according to your above comments.

Response:

We have checked and revised the above English writing problems in manuscript according to your above comments.

---

## Author Comment (AC3)

**General Comments**

This paper presents an in-situ observational dataset on permafrost thermal regimes along the China-Russia Crude Oil Pipeline (CRCOP) route in northeast China, consisting of meteorological observations, soil temperature and soil moisture content data, and electrical resistivity tomography (ERT) data. The analysis of this dataset shows that the operation of CRCOP has had profound effects on the thermal state of the ground. The results are useful for better understanding the responses of permafrost to climate change and engineering activities. This dataset is valuable for evaluating the integrity of pipelines and the effectiveness of measures to mitigate the permafrost thaw, as well as for validating numerical models. The manuscript is overall well organized and written, but there are still some shortcomings that need to be addressed.

**Specific comments**

1, Please enhance the description of data processing (e.g. how do you deal with missing data and how these missing data affect the results?) in the revised manuscript. This will be important for full understanding of this dataset.

Response:
Thank you for your suggestion. We have added the description of quality control of data in the revised manuscript. Obvious erroneous recordings are manually removed and all the missing or abnormal data are replaced with null values before daily average values are calculated.

2, Warm-oil pipeline dissipates heat into the surrounding permafrost, resulting in thermal and physical disturbances to the pipeline right-of-way. These disturbances can compromise pipeline integrity and pose the potential risk of oil leakage. Therefore, in-situ permafrost monitoring has been made along several important pipeline routes (e.g. the Norman Wells pipeline and Trans-Alaska oil pipeline), where reliable first-hand data has been collected. I suggest the authors mention those important studies as background to this study.

Response:
What you mentioned above is right. Pipelines constructed in permafrost are inevitably faced to major challenges related to thaw settlement, frost heave, slope failure, icing, and frost mounds. Therefore, the monitoring systems were established along several pipeline routes, such as the Norman Wells pipeline and the Trans-Alaska oil pipeline, to obtain field data for understanding how the permafrost foundation performed when a pipeline went through or above it. Such datasets and associated analyses are valuable because they can provide references and implications for CRCOP. According to your suggestion, these important studies have been added to the introduction section in red in the revised manuscript as follows:
"As a result, a permafrost monitoring network along the CRCOPs route was gradually established by referring to the experiences and lessons from other oil/gas pipelines (e.g. Norman Wells to Zama oil pipeline in Canada, Alyeska oil pipeline in the U.S., and Nadym–Pur–Taz natural gas pipeline in Russia) in permafrost regions (Burgess and Smith, 2003; Johnson and Hegdal, 2008; Smith and Riseborough, 2010; Oswell, 2011). Boreholes were instrumented to measure GTs in the

active layer and near-surface permafrost on and off the right-of-way (ROW) of the CRCOPs and electrical resistivity tomography (ERT) surveys were used to delineate frozen and unfrozen ground in the vicinity of the CRCOPs (Kneisel et al., 2008; Farzamian et al., 2020) ".

Newly added references as follows:
*Burgess M M, Smith S L. 17 years of thaw penetration and surface settlement observations in permafrost terrain along the Norman Wells pipeline, Northwest Territories, Canada. Proceedings of the Eighth International Conference on Permafrost, 2003: 107-112.*
*Johnson E R, Hegdal L A. Permafrost-related performance of the Trans-Alaska oil pipeline[C]. Proceedings of 9th International Conference on Permafrost, Fairbanks, AK, USA. 2008: 857-864.*
*Smith S L, Riseborough D W. Modelling the thermal response of permafrost terrain to right-of-way disturbance and climate warming[J]. Cold Regions Science and Technology, 2010, 60(1): 92-103.*
*Oswell J M. Pipelines in permafrost: geotechnical issues and lessons[J]. Canadian Geotechnical Journal, 2011, 48(9): 1412-1431.*

3, Section 2. I found the locations of the sites are not accurate enough, rounded to two decimal places. The locations of boreholes and ERT profiles are not given in the manuscript, nor in the associated dataset. I suggest the author can provide accurate locations (at least four decimal places in unit degrees) for these mentioned locations.

Response:
Thank you for your suggestions. We have provided accurate locations for the five monitoring sites in Table 1 with four decimal places in unit degrees. The boreholes and ERT profiles have been given ID according to where they exist. For example, for JB-B-1 the -B- indicates that is a 'borehole', and the prefix JB is an abbreviation for the site name. So, the locations of boreholes can be determined by their name prefixes.

4, Ground temperature was automatically collected by the dataloggers of RTB37a36V3 and CR3000 or measured manually with. Please provide a description of the errors occurring in these measurements.

Response:
Results show that the accuracy of the manual readings is estimated to be ±0.1 °C, which is lower than the RTB37a36V3 and CR3000 data loggers because the data measured with Fluke 87/89 was collected once at an instantaneous time (Juliussen et al., 2010). However, there are no overlapping data collected via these three methods at the same borehole, we can not quantify the deviation of the three different recording methods.
Reference:
*Juliussen H, Christiansen H H, Strand G S, et al. NORPERM, the Norwegian permafrost database–a TSP NORWAY IPY legacy[J]. Earth System Science Data, 2010, 2(2): 235-246.*

5, The ROW widths are equal at each monitoring site? Please give a clear description.

Response:

Generally, the ROW is approximately 20 m wide along the pipeline route except for some special sites, such as poor geological conditions. In the revised manuscript, we added the description.

6, Figure 3, seems problematic in the caption. According to the caption, column (a) indicates the active layer, but ground temperatures in XT in (a2-4) were measured below zero degrees for several consecutive years, which actually implies permafrost at this depth.

Response:
We are sorry for this mistake. The caption was revised as "Variability of ground temperatures at depths of 0–3 m (a) and 8–20 m (b) at the undisturbed sites along the route of CRCOPs in Northeast China, 2018–2021.".

7, In Lines 110-111, please give exact timing for the borehole drilling.

Response:
The exact timing has been added in the manuscript, as described as "Besides, a new borehole (JB-B-I) was drilled in March 2017 down to 60.6 m near the above-mentioned AWS.".

8, Line 232 to 240, ERT results show that a talik formed around CRCOP I is much larger than that around CRCOP II. What is the reason?

Response:
The CRCOP I was constructed during 2009-2010 and began to operate in 2011. To reach the requirement of 30-million-ton throughput per year signed by the governments of China and Russia, an equal-size new pipeline, i.e. CRCOP-II, in parallel with CRCOP-I was built in 2017 and began operation in 2018. Due to a longer operation time and heating from the CRCOP I, the talik formed around CRCOP I is much larger than that around CRCOP II.

9, Line 255-257, it's difficult to directly observe the 1.5 m cooling range of two-phase closed thermosyphons and the 4 m lateral extent of thermal disturbance in Fig.9c and d, please clarify this point.

Response:
Thank you for your suggestion. Difficulties in understanding and even misunderstandings may arise if the distances of boreholes and thermosyphons from the pipe centerline were not provided. In the revised manuscript, this detailed information has been added in Table 3.

10, In Figure 8 and Appendix D, please use blue for higher resistivity and red for lower resistivity in ER images, like the color scheme in Figure 9. This is more intuitive for readers.

Response:
We re-draw Figure 8 and Appendix D according to your suggestion.

11, In Figure 10, please add the unit for soil volumetric liquid water content.

Response:

Yes, we added the unit in Figure 9 in the revised version.

---

## Author Response (AR1)

**Response Letter**

We thank the reviewers very much for their kind comments and constructive suggestions regarding the revisions of this manuscript. We have carefully considered all comments and suggestions and made a thorough revision to the original manuscript based on them. All ensued followups are also made. Our point-by-point responses to the comments are listed below in blue colour.

Sincerely,
Guoyu Li, Huijun Jin and Fei Wang

**Reviewer 1:**

**1.1 General Comments**

The paper submitted by Li et al. describes a permafrost dataset compiled from measurements made along an oil pipeline route in NE China. The dataset includes information on ground temperatures, soil moisture along with results of geophysical surveys and meteorological data. Instrumented sites are located at various distances from the pipeline and are both on the right-of-way (ROW) and in undisturbed terrain off the ROW which facilitates characterization of impacts of pipeline characterization, similar to that done along other pipelines (e.g. Burgess and Smith 2003; Burgess et al. 2010; Johnson and Hegdel 2008). Such datasets and associated analysis are valuable because they can be utilized for model calibration and validation and also to evaluate mitigation techniques to reduce impacts and improve design of future infrastructure projects (e.g. Burgess et al. 2010). For these reasons, the paper and associated database should of interest to engineers and those involved in environmental impact assessment. However the manuscript does require a number of revisions before it can be accepted for publication.

As mentioned above, this is not the first time monitoring programs have been established along pipeline routes in permafrost regions and the data utilized to evaluation design performance, effectiveness of mitigation techniques as well as the lessons learned being used for design of other projects. For example, there are several papers and reports on two pipeline corridors in NW North America (Norman Wells to Zama (NWZ) oil pipeline NWT Canada and Alyeska oil pipeline in Alaska). These include for example Burgess et al. (2010); Burgess and Smith (2003); Smith and Burgess (2010); Croft et al. (2021); Mosely et al. (2021); Johnson and Hegdel (2008), as well as other reports referenced in these. Data compilations are available for the NWZ pipeline (Smith et al. 2004, 2008) and may also be available for the Alyeska oil pipeline. It would be useful to refer to these other studies in both the introduction to provide the rationale for compilation of similar databases and also when interpreting the results of data analysis.

Thank you for the comment. Pipelines constructed in permafrost have inevitably faced to many problems related to frost heave, thaw settlement, slope failure, icing, and frost mounds, which were forced to establish monitoring systems for permafrost environmental and stability evaluation

such as along the Norman Wells pipeline and the Trans-Alaska oil pipeline, and to obtain field data for understanding how the permafrost foundation performed when pipelines went through or above it. Correspondingly. Such datasets and associated analyses are valuable for this manuscript because they do provide references and implications for CRCOPs when preparing this manuscript. According to your comment, most of these studies have been introduced and referred to in this revised manuscript as follows:

*'As a result, a permafrost monitoring network along the CRCOPs route was gradually established by referring to the experiences and lessons learned from other oil and gas pipelines (e.g., Norman Wells to Zama crude oil pipeline in Canada, Alyeska crude oil pipeline in the U.S., and Nadym–Pur–Taz natural gas pipeline in Russia) in permafrost regions (Burgess and Smith, 2003; Johnson and Hegdal, 2008; Smith and Riseborough, 2010; Oswell, 2011). Boreholes were instrumented to measure GTs in the active layer and near-surface permafrost on and off the right-of-way (ROW) of the CRCOPs and electrical resistivity tomography (ERT) surveys were used to delineate frozen and unfrozen ground in the vicinity of the CRCOPs (Kneisel et al., 2008; Farzamian et al., 2020).'.*

Newly added references as follows:

*Burgess M M, Smith S L. 17 years of thaw penetration and surface settlement observations in permafrost terrain along the Norman Wells pipeline, Northwest Territories, Canada. Proceedings of the Eighth International Conference on Permafrost, 2003: 107-112.*
*Johnson E R, Hegdal L A. Permafrost-related performance of the Trans-Alaska oil pipeline. Proceedings of Ninth International Conference on Permafrost, Fairbanks, AK, USA. 2008: 857-864.*
*Smith S L, Riseborough D W. Modelling the thermal response of permafrost terrain to right-of-way disturbance and climate warming. Cold Regions Science and Technology, 2010, 60(1): 92-103.*
*Oswell J M. Pipelines in permafrost: geotechnical issues and lessons. Canadian Geotechnical Journal, 2011, 48(9): 1412-1431.*

The paper could be better organized and some important information should be provided. In particular, Section 2 (Study site and instrumentation) could be better written and include some information from Section 3 (Data description). It would be better to clearly describe the instrumentation used to measure each variable (i.e. ground temperature, moisture content, air temperature etc.) and use a table to indicate which variables are measured at each site/borehole. This would reduce repetition and enhance clarity. Some of the information in Section 3 also describes instrumentation and should be moved to Section 2 so that all methods appear in the same section.

Thanks for pointing out this issue. In the revised manuscript, Section 2 (Site description) and Section 3 (Data description) have been re-organized by referring to your comments and the recently published papers in the ESSD (e.g. Boike et al., 2019; Wu et al., 2022). Section 3 is divided into three sub-sections: meteorological data, ground temperature and soil water content data, and subsurface electrical resistivity data.

Information on distance of boreholes from the pipe centre line for those on the ROW needs to be clearly provided. Also, the ROW width at each site should be provided as well as the distance from the ROW edge for boreholes that are in the undisturbed terrain off the ROW. This

information is important as it provides information on the amount of disturbance there might be at each site. This information could be provided in a table or as site plans in the supplementary information.

Thanks for your better suggestions. We are sorry that we did not provide the exact locations of boreholes and the pipeline ROW width. The ROW width of the CRCOPs is approximately 20 m along its route. The distances of boreholes from the pipe centerline of CRCOP I have been added in Table 3 in the revised manuscript as follows. Four boreholes of JS-B-1, JS-B-2, SL-B-1, and SL-B-2 were drilled on the ROW of CRCOP II. Their distances from the centerline of CRCOP II are provided. In turn, the distances of off-ROW boreholes from the ROW edge are easily determined.

Table 3 Summary of monitoring information of ground temperature boreholes, water content monitoring pits, and electrical resistivity tomography (ERT) profiles along the China-Russia Crude Oil Pipelines (CRCOPs) in Northeast China.

| Variable | Borehole/ ERT profile | Maximum monitoring depth (m) | Distance from pipe centreline (m) | Data logger | Measuring internal | Operation period |
|---|---|---|---|---|---|---|
| Soil/permafrost temperature at the natural site (off-ROW) | XA-B-I | 60.5 | 100 | CR 3000 | 2h, AUTO | Nov 2018 – Nov 2020 |
| | XT-B-I | 20 | 80 | RTB37a36V3 | 2h, AUTO | Jul 2019 – Aug 2021 |
| | JS-B-I | 20 | 24.8 | CR 3000 | 2h, AUTO | Dec 2018 – Jun 2021 |
| | SL-B-I | 25 | 12.6 | Fluke 87/89 RTB37a36V3 | Monthly, MANU 2h, AUTO | Sep 2017 – Oct 2019 Aug 2020 – Dec 2020 |
| | JB-B-I | 60.6 | 80 | CR 3000 | 2h, AUTO | Jun 2018 – Aug 2020 |
| | JB-B-II | 20 | 16.6 | Fluke 87/89 CR 3000 | Monthly, MANU 2h, AUTO | Nov 2011 – Sep 2017 Oct 2017 – Aug 2021 |
| Soil/permafrost temperature on pipeline ROW (on-ROW) | JS-B-1* | 19.8 | 2 | CR 3000 | 2h, AUTO | Oct 2017 – May 2021 |
| | JS-B-2* | 20 | 5 | CR 3000 | 2h, AUTO | Oct 2017 – Aug 2021 |
| | SL-B-1* | 24.8 | 3 | Fluke 87/89 RTB37a36V3 | Monthly, MANU 2h, AUTO | Sep 2017 – Oct 2019 Aug 2020 – May 2021 |
| | SL-B-2* | 24.8 | 5.9 | Fluke 87/89 RTB37a36V3 | Monthly, MANU 2h, AUTO | Sep 2017 – Oct 2019 Aug 2020 – May 2021 |
| | JB-B-1 | 20 | 2 | Fluke 87/89 CR 3000 | Monthly, MANU 2h, AUTO | Mar 2014 – Sep 2017 Oct 2017 – Aug 2021 |
| | JB-B-2 | 15 | 2 | CR 3000 | 2h, AUTO | Jun 2015 – Aug 2021 |
| | JB-B-3 | 15 | 2 | | 2h, AUTO | Jun 2015 – May 2018 |
| | JB-B-4 | 15 | 2 | | 2h, AUTO | Jun 2015 – May 2019 |
| | JB-B-5 | 10 | 3 | | 2h, AUTO | Jun 2015 – Aug 2021 |
| | JB-B-6 | 14 | 3 | | 2h, AUTO | Jun 2015 – May 2018 |
| | JB-B-7 | 15 | 3 | | 2h, AUTO | Jun 2015 – May 2020 |
| | JB-B-8 | 15 | 4 | | 2h, AUTO | Jun 2015 – Aug 2021 |
| | JB-B-9 | 15 | 4 | | 2h, AUTO | Jun 2015 – May 2018 |
| | JB-B-10 | 15 | 4 | | 2h, AUTO | Jun 2015 – May 2020 |
| Soil volumetric liquid water content on pipeline ROW | JB-W1 JB-W2 JB-W3 | 2.5 | 1 | CR 3000 | 2h, AUTO | Jun 2015–Aug 2021 |
| Electrical resistivity | P-JS | 24 | | SuperSting R8 system | Site visit | Apr 11, 2018 |
| | P-SL | 24 | | | Site visit | Apr 12, 2018 |
| | P-JB-1 | 24 | | | Site visit | Apr 06 – Apr 10, 2018 |

| | |
|---|---|
| P-JB-2 | 24 |
| P-JB-3 | 24 |
| P-JB-4 | 18 |

\* Boreholes were drilled on the ROW of CRCOP II. The width of ROW along the pipeline is about 20 m.

Section 3 doesn't really provide a clear description of the data in the dataset but rather provides some information on instrumentation as well as presentation of some of the data in graphic form and interpretation of the results. It would be useful to have a brief and clear description of the data. For example, are hourly, daily, annual mean values provided; if annual values provided, what is the time period (calendar or hydrologic year) and how do you deal with missing values; describe any processing done to ensure data quality and any derived parameters that might be included in the dataset. The presentation of the data and interpretation of results could be in a separate section.

Section 3 has been revised to improve the description of the data according to your comments. We have added the description of quality control of data in the revised manuscript. Obvious erroneous recordings are manually removed and all the missing or abnormal data are replaced with null values before daily average values are calculated. All meteorological data from the small automated AWS, GT data at different depths in boreholes, and VWC data, at the daily resolution, have been compiled in this dataset.

A number of specific comments are provided below. Some of these are editorial while others are related to clarity of the content. I hope the authors find the comments helpful in preparing an improved manuscript that should be acceptable for publication.

We respond to all of your comments below.

**1.2 Specific Comments**

L19-20 – suggested revision: "In 2011 we initiated a…."

It was revised according to your suggestion.

L21-24 – Not clear. Do you mean you compiled an integrated dataset on the ground thermal state utilizing the various types of data that are mentioned in the first part of the sentence. You should revise to be clear.

The sentence has been revised as

'*We compiled an integrated dataset of the ground thermal state along the CRCOPs route, consisting of meteorological data near the southern limit of latitudinal permafrost, ground temperature data in 20 boreholes with depths of 10.0–60.6 m, soil volumetric liquid water contents and 2-dimensional electrical resistivity tomography (ERT) data at different sites.*'.

L24–26 – Are you referring to the undisturbed sites off the ROW? Revise to be clear. A revision is also suggested: "Results demonstrate that permafrost has warmed between 2011-2020 in the vicinity of SLLP, manifested as an increase in ground temperature at depth likely in response to climate change."

Thanks. The sentence has been revised according to your suggestion.

L28 – Insert "an: between "as" and "insulation"

Added as suggested.

L37-38 – I suggest you remove the older references (ones between 1999 and 2008) and add more recent ones such as Etzelmuller et al. (2020); Zhao et al. (2020) and Liu et al. (2021). You could also cite the most recent BAMS State of the Climate sections on permafrost (i.e. Smith et al. 2021 and Noetzli et al. 2021).

L39 – You could delete the older Zhang et al. ref and add newer ones such as Burke et al. (2020) and other recent papers (you could also cite most recent IPCC reports of 6th Assessment).

L41 – I think you can delete the older references here (before 2010) as you have several recent ones.

According to your suggestions, the older references have been deleted and more recent references have been added. The sentence has been revised as

'*Over the last few decades, the warming and thawing of permafrost have been observed in most permafrost regions (e.g., Ran et al., 2018; Biskaborn et al., 2019; O'Neill et al., 2019; Etzelmüller et al., 2020; Liu et al., 2021; Noetzli et al., 2021; Smith et al., 2022), and permafrost degradation will continue in response to a warming climate (Koven et al., 2013; Burke et al., 2020). Permafrost degradation affects the geomorphological characteristics, carbon release, hydrological process, ecosystem, climate system, and integrity of infrastructure (Cheng and Jin, 2013; Beck et al., 2015; Hjort et al., 2018, 2022; Turetsky et al., 2020; Jin and Ma, 2021; Jin et al., 2021, 2022; Luo et al., 2021; Jones et al., 2022; Liu et al., 2022; Miner et al., 2022)*'.

L70-71 – Permafrost presence/absence is inferred or interpreted from the ERT results rather than "detected". A revision is suggested: "… electrical resistivity tomography (ERT) surveys were used to determine the distribution of frozen and unfrozen ground in the vicinity of the CCOPs……ERT surveys to characterize freeze-thaw dynamics in…."

We are sorry for this unreasonable description. The ERT method is commonly used in measuring the electrical resistivity distribution in the subsurface and has proved to be a useful geophysical tool for obtaining information about the active-layer configuration, permafrost thickness, and the interfaces between frozen and unfrozen grounds in permafrost studies. The sentence has been revised to avoid the existing misleading information according to the suggestion and the above general comments:
'*... electrical resistivity tomography (ERT) surveys were used to delineate frozen and unfrozen ground in the vicinity of the CRCOPs (Kneisel et al., 2008; Farzamian et al., 2020).*'.

L68-74 – I don't think you need this much detail here as it should be covered in the methods section (section 2). You could just say that boreholes were instrumented to measure ground temperatures and ERT surveys were used to delineate frozen and unfrozen ground.

Thanks. The sentence has been revised as

'*Boreholes were instrumented to measure GTs in the active layer and near-surface permafrost on and off the right-of-way (ROW) of the CRCOPs and electrical resistivity tomography (ERT) surveys were used to delineate frozen and unfrozen ground in the vicinity of the CRCOPs (Kneisel et al., 2008; Farzamian et al., 2020).*'.

L82 – replace "are" with "were"

It was revised.

L88-90 – Revision suggested "Between 1972 and 2017, MAAT increased at a …" You can also delete "climate wetting".

According to the suggestion, the sentence has been revised as

'*Between 1972 and 2017, MAAT increased at a rate of 0.32 °C per decade while annual precipitation increased at a rate of 14.6 mm per decade (Wang et al., 2019a).*'.

L91 – You can delete the last part of the sentence from "retrieved" onward. The additional information can be provided in the Table (or its caption) if you need to provide references to previous studies.

It was deleted accordingly.

L92-93 – Suggested revision – "Permafrost is warm with mean annual ground temperature…….ranging from -1.8 to -0.4°C. The reference may not be necessary as it seems to be a textbook, rather than a reference that specifically reports on MAGT in the region – perhaps one of the other papers in the list would be more suitable.

The sentence has been revised as

'*Permafrost is warm with mean annual ground temperature (MAGT) at the depth of zero annual amplitude (DZAA) ranging from −1.8 to −0.4 °C.*'

according to the suggestion, and the less related reference was deleted.

L95-97 – Suggested revision – "The XA site, located in a permafrost wetland, is the most northern site and has the lowest air temperature, while……China has the highest air temperature and permafrost occurs in isolated patches.

The sentence has been revised according to the suggestion.

L99-100 – Can't you just say it is ice-rich permafrost?

Yes, it was revised as 'ice-rich permafrost'.

L108 – replace "built" with "installed"

It was revised accordingly.

L111 – You could just say when the hole was drilled and instrumentation installed. It isn't clear why it is important to say that it was finished 6 months before installation of AWS. If it was a long drilling time and likely caused significant disturbance to the thermal regime just give the number of days of drilling and the implication of this.

We are sorry for this vague statement. The sentence has been revised as

'*Besides, a new borehole (JB-B-I) was drilled in March 2017 down to 60.6 m near the above-mentioned AWS.*'.

L119 – How far off the ROW were these sites – this information should be provided.

The distances of off-ROW boreholes at each site from the centerline of CRCOP I were added in Table 3 in the revised manuscript.

L118-139 – Some of the detail could be reduced by putting some of the information in a table (or figure) such as the distance from the pipe, or depth of borehole. The table could also summarize all the instrumentation or surveys at a site, i.e. add this information to Table 3 for example. In this section you could just mention the instrumentation used to acquire information on the various parameters such as GT, soil moisture etc. You should also indicate how wide the ROW is at each site as this will give the reader an indication of the amount of the disturbance.

This paragraph has been rewritten and some necessary information has been added in Table 3. Details can be found in the above responses to the comment.

L122-123 – Don't you mean the boreholes were drilled vertically? (rearrange sentence)

Sorry for the unclear statement. Yes, all boreholes were drilled vertically. This sentence has been revised as

'*At the SL site, two on-ROW boreholes (SL-B-1 and SL-B-2, 3 and 5.9 m from the centerline of CRCOP II, respectively) and one off-ROW borehole (SL-B-I, 2.6 m from the edge of the CRCOP I ROW) were drilled in March 2017 and instrumented in September 2017.*'.

L129 – Weren't thermistor cables used in other BHs?

The GT measurement was carried out by installing a thermistor cable protected by a steel tube into the borehole. When a borehole was completed, a steel tube was installed in it. Then, the borehole outside the steel tube was densely backfilled with fine-grained sand and a thermistor cable was placed in the steel tube. Thermistor cables containing thermistors at designed intervals were used to monitor ground temperatures.

L134 – Do you mean precision (resolution) rather accuracy here? Wouldn't accuracy be more related to the temperature sensor?

Agree. In the revised version, we changed this sentence to

'*The latter generally has a lower resolution than the CR3000 data logger (±0.05℃), but allows more widely used due to its lower cost*.'.

L148 Table 2 – The resolution/precision of instrumentation, including the dataloggers used should also be provided.

The accuracy of the manual readings is estimated to be ±0.1 ℃, which is lower than the CR3000 data logger (±0.05℃) (Juliussen et al., 2010). The GT data collected by the RTB37a36V3 data logger is not as steady as that collected by the CR3000 data logger, i.e., the RTB37a36V3 data logger has a lower resolution than the CR3000 data logger. This content has been added to the revised manuscript.
*Juliussen H, Christiansen H H, Strand G S, et al. NORPERM, the Norwegian permafrost database–a TSP NORWAY IPY legacy. Earth System Science Data, 2010, 2(2): 235-246.*

L151 – Table 3 – You could just give the maximum measurement depth in the table as the various measurement depths would be included in the database. Alternatively, you could include this additional information as a supplementary information table. You could then include the additional information in the table suggested in an earlier comment (distance from pipe, ROW width at each site, and whether BH is on or off ROW).

Table 3 has been improved and revised according to several constructive comments.

L153 – Section 3 - Data description – You seem to be confusing methodology with dataset description. Much of this information including type of instrumentation used should be included in Section 2 which describes methods including instrumentation. In some cases such as Section 3.2 there is a repeat of the instrumentation description that was given in Section 2. Section 3 also presents results, analysis and discussion so some consideration should be given to revising the organization of sections.

According to your suggestions, the paper was reorganised and Section 3 has been revised to improve the description of the data. Details can be found in the above responses.

L190 – Fig. 3b doesn't clearly show the relationship between GT and latitude and whether it is linear or not. MAGT at DZAA for each undisturbed site could be plotted vs latitude to better show this.

As shown in Figure 3b, the annual changes of ground temperatures at the depth of 15m at all monitoring sites are negligible except for the JS site. This depth is considered as the depth of zero annual amplitude (DZZA). The MAGTs at DZAA at the sites of JB, SL, XT, and XA are −0.7, −0.8, −1.8, and −1.7°C respectively, indicating that the MAGT at DZAA is greatly affected by the latitude, but the relationship between MAGT and latitude is not linearly dependent, as shown in Figure below, which was added in the revised manuscript and numbered Figure 4.

[Figure]

Figure 4. Relationship between latitude and GT along the route of China-Russia Crude Oil Pipelines (CRCOPs).

L192-193 – Are you sure this isn't an issue with the sensors?

We think it is not the failure of sensors. Firstly, the thermistor cable used to measure GT has a high accuracy of ± 0.05 °C and the measuring range of the thermistor ranges from −30 to 30 °C. The collected data thus are reliable. Secondly, this abnormal phenomenon has occurred at depths of 10, 15, and 20 m for two consecutive years, which would seem to indicate that the failure of the

thermistors is impossible. In addition, the strata of this site (obtained from the JS-B-I borehole) are composed of peaty soil (0-0.8 m), grey silty clay (0.8-2.0 m), yellowish-brown gravel (2.0-5.5 m), and strong weathered granite (5.5-20.0 m). The presence of the rock layers with high permeability provides a channel for the movement of intra-permafrost water.

L197-199 – A bit confusing as Fig. 4 shows seasonal variation.

It was changed in the revised manuscript.

L199-200 – Are you referring to the thermal offset here?

We describe the characteristics of the ground temperature curve. As for the thermal offset, it is not discussed. At depths from 1 to 2 m, the monthly average GTs in 2018 was fluctuating in proximity to 0 °C.

L201-204 – A couple of other things to consider: ALT also responds more to shorter-term variations in air temperature compared to the deeper ground temperatures for which the higher frequency variations are filtered out. ALT will therefore show more interannual variability as shown in Fig. 5. The ground at JB is ice-rich, and melting of ground ice as thaw progresses deeper into the ground can lead to surface subsidence and consolidation of the unfrozen material. Changes in ALT over time can be relatively small as ice-rich material thaws compared to sites where excess ice is negligible. See for example Nyland et al. (2021), O'Neill et al. (2019).
L217-218 – The key thing is that insulation doesn't prevent heat transfer but reduces it.

Thanks for your comments. The above questions have been considered carefully and interpreted in revising the manuscript. Meanwhile, the relevant references are added.

L219-221 – As well as the effect of the pipe, any ROW disturbance such as changes to the surface, clearing of vegetation will also have an effect on the ground thermal regime. These changes could also effect the ground thermal regime off ROW (lateral heat transfer). See for example some of the publications on the Norman Wells pipeline mentioned earlier and also Burgess and Smith (2003), Smith and Riseborough (2010).

Agree, our results also show that many factors lead to permafrost thawing and thaw settlement along the pipeline route including ROW vegetation clearing, trenching, warm oil temperature, surface water infiltration, groundwater flowing, and climate warming, and the thermal effects of ROW disturbance can extend to the adjacent undisturbed terrain, which is mainly due to small separation distance between two parallel pipelines and increasing oil temperature. The related reference has been added in the revised manuscript.

L225 – It isn't clear why this is an artificial permafrost table – is this the site with thermosyphons? If so, it should be mentioned first.

We are sorry for this mistake. It was changed to the permafrost table. Meanwhile, this paragraph has been rewritten and some necessary information has been added in Figure 8.

'*The time-series of the depths of the permafrost table and maximum frost penetration in borehole JB-B-1, horizontally 2 m away from the uninsulated pipe centerline at the JB site shows that since the official operation of CRCOP I starting in 2011, the depth of the permafrost table has been increasing with an average rate of 0.9 m yr-1 and depth of seasonal frost penetration decrease*

*rapidly and then varies little (0.4-0.8 m) during 2010–2021 (Fig. 8). Therefore, a thawed interlayer between permafrost table and the bottom of seasonal frost (i.e., supra-permafrost subaerial talik, SST) formed and developed with an average rate of 1.1 m yr-1 in the same period.'.*

[Figure]

Figure 8. Variations in the depths of permafrost table and seasonal frost and thickness of supra-permafrost subaerial talik during 2010–2021 in borehole JB-B-1, 2 m away from the centerline of the uninsulated China-Russia Crude Oil Pipeline (CRCOP) I at the JB site in northern part of Northeast China.

L228-230 – Are you referring to surface settlement (subsidence) here?

Here it refers to the total settlement of pipeline foundation soil, and it was revised.

L245 – Figure 8 – Consider reversing the colour scale and using blue for higher resistivity (colder ground –permafrost) and red for lower resistivity (warmer ground – unfrozen). This is more intuitive and has been done in other papers presenting ERT results. If some of the transect is off ROW, indicate the ROW edge. Was there a topographic survey done? – no change in ground elevation is shown.

Thank you for your comments. The ERT surveys have been conducted at JB, SL, and JS sites. These three sites are selected on relatively flat terrain and there are no obvious changes in ground elevation. Thus, the ground surface varies little in elevation in RET images. The color for ERT images has been changed in this figure and appendix D based on your suggestion. The updated figures are shown below:

[Figure]

Revised as Figure 11 in the new version.

[Figure]

Figures D1, D2, and D3.

L246 – There acronym TPCT isn't really needed – just say thermosyphons in the rest of the paragraph.

It has been changed.

L246-262 – The papers mentioned earlier on the Alaska Alyeska oil pipeline might be relevant here as the effect of thermosyphons are described.

Yes, the relevant references have been added.

L269 – Volumetric moisture content doesn't really have units. (dimensionless) You could give as %.

Yes, we added the unit in Figure 9 in the revised version.

**References**

Burgess MM, Oswell J, Smith SL 2010. Government-industry collaborative monitoring of a pipeline in permafrost – the Norman Wells Pipeline experience, Canada. In: GEO2010, 63rd Canadian Geotechnical Conference and the 6th Canadian Permafrost Conference, Calgary, Sept 2010. GEO2010 Calgary Organizing Committee, pp 579-586. https://www.aina.ucalgary.ca/scripts/mwimain.dll/116/2/3/72659?RECORD&DATABASE=CPC

Burgess MM, Smith SL 2003. 17 years of thaw penetration and surface settlement observations in permafrost terrain along the Norman Wells pipeline, Northwest Territories, Canada. In: Phillips M, Springman SM, Arenson LU (eds) Proceedings of 8th International Conference on Permafrost, Zurich Switzerland, July 2003. A.A. Balkema, pp 107-112

Burke EJ, Zhang Y, Krinner G (2020) Evaluating permafrost physics in the Coupled Model Intercomparison Project 6 (CMIP6) models and their sensitivity to climate change. The Cryosphere 14:3155-3174. doi:10.5194/tc-14-3155-2020

Croft PE et al. 2021. Slope Stabilization along a Buried Crude-Oil Pipeline in Ice-Rich Permafrost. Proceedings of The Regional Conference on Permafrost 2021 and the 19th International Conference on Cold Regions Engineering, American Society of Civil Engineers. p. 339-350 https://doi.org/10.1061/9780784483589

Etzelmüller, B. et al. (2020). Twenty years of European mountain permafrost dynamics — the PACE legacy. Environ. Res. Lett. 15, 104070.

Johnson ER and Hegdel LA 2008. Permafrost related performance of the Trans Alaska oil pipeline. Proc. 9th Int. Conf. on Permafrost, Fairbanks Alaska. 857-864

Liu et al. 2021. Permafrost warming near the northern limit of permafrost on the Qinghai–Tibetan Plateau during the period from 2005 to 2017: A case study in the Xidatan area. Permafrost and Periglacial Processes 32:323–334. https://doi.org/10.1002/ppp.2089

Mosley, L et al. 2021. Alyeska's 40-Plus Years of Experience with Heat Pipes on the Trans-Alaska Pipeline System. Proceedings of The Regional Conference on Permafrost 2021 and the 19th International Conference on Cold Regions Engineering, American Society of Civil Engineers. p. 327-338 https://doi.org/10.1061/9780784483589.031

Noetzli J, Christiansen HH, Hrbáĉek F, Isaksen K, Smith SL, Zhao L, Streletskiy DA (2021) [Global Climate] Permafrost Thermal State [in "State of the Climate in 2020"]. Bulletin of the American Meteorological Society 102 (8):S42-S44. doi:10.1175/BAMS-D-21-0098.1

Nyland, K.E., Shiklomanov, N.I., Streletskiy, D.A., Nelson, F.E., Klene, A.E., Kholodov, A.L., 2021. Long-term Circumpolar Active Layer Monitoring (CALM) program observations in Northern Alaskan tundra. Polar Geogr. 44, 167–185. https://doi.org/10.1080/1088937X.2021.1988000

O'Neill, H.B., Smith, S.L., Duchesne, C., 2019a. Long-term permafrost degradation and thermokarst subsidence in the Mackenzie Delta area indicated by thaw tube measurements, in: Cold Regions Engineering 2019. American Society of Civil Engineers, Reston, VA, pp. 643–651. doi:10.1061/9780784482599

Smith SL, Romanovsky VE, Isaksen K, Nyland KE, Kholodov AL, Shiklomanov NI, Streletskiy DA, Farquharson LM, Drozdov DS, Malkova GV, Christiansen HH (2021) [Arctic] Permafrost [in "State of the Climate in 2020"]. Bulletin of the American Meteorological Society 102 (8):S293-S297. doi:10.1175/BAMS-D-21-0086.1

Smith SL, Burgess MM 2010. Long-term field observations of cyclical and cumulative pipe and ground movements in permafrost terrain, Norman Wells Pipeline, Northwest Territories Canada. In: GEO2010, 63rd Canadian Geotechnical Conference and the 6th Canadian Permafrost Conference, Calgary, Sept. 2010. GEO2010 Calgary Organizing Committee, pp 595 https://www.aina.ucalgary.ca/scripts/mwimain.dll/116/2/4/72661?RECORD&DATABASE=CPC-602

Smith SL, Riseborough DW (2010) Modelling the thermal response of permafrost terrain to right-of-way disturbance and climate warming. Cold Regions Science and Technology 60 (1):92-103

Smith SL, Burgess MM, Riseborough D, Chartrand J (2008) Permafrost and terrain research and monitoring sites of the Norman Wells to Zama pipeline – Thermal data collection and case histories, April 1985 to September 2001. Geological Survey of Canada Open File 5331. doi:10.4095/224831

Smith SL, Burgess MM, Riseborough D, Coultish T, Chartrand J (2004) Digital Summary Database of Permafrost and Thermal Conditions – Norman Wells Pipeline Study Sites. Geological Survey of Canada Open File 4635. doi:10.4095/215482

Zhao, L. et al. (2020) Changing climate and the permafrost environment on the Qinghai–Tibet (Xizang) plateau. Permafrost and Periglac Process 31, 396–405.

**Reviewer 2:**

**2.1 General Comments**

This paper presents an integrated dataset on the permafrost along the China Russia crude oil pipeline in northeast China, which is an important complement to the current global permafrost database. These data are of great significance to the stability of linear infrastructures (e.g., pipeline, highway, railway), the validation of numerical simulation models, as well as the prediction of permafrost evolution process (degradation or aggradation). However, the manuscript still needs some revisions to reach a publishable level. My detailed comments as well as some suggestions are listed as below. Hope these comments will be useful to the authors to improve the manuscript.

Thank you very much for your comments. We have made a thorough revision to the original manuscript based on your comments and suggestions.

**2.2 Specific Comments**

Line 35: It should be better to say "the Earth's cryosphere".
Line 35-36, rewrite, "the thermal state" of permafrost" is not a component of cryosphere
Line 67: State Key Laboratory of Frozen Soil Engineering (SKLFSE), not soils. Check
Line 69: Delete the definite article "the" before electrical resistivity tomography (ERT).
Line 76: You can delete the comma before "in Northeast China", or keep the comma, but delete "in" before Northeast China.
Line 82: "…… were established ……"
Line 84: "…… was primarily based on ……"
Line 90: Delete "climate wetting at".
Line 108: "…… was installed ……"

Thanks for the suggestions, these sentences have been revised accordingly and highlighted in red in the revised manuscript.

Line 110-111: The sentence in the parenthesis is confusing. Do you mean the borehole was drilled 6 months before the installation of AWS? If so, you can just give the exact date when the bole was drilled to avoid any confusion.

We are sorry for this unclear statement. The sentence has been revised as

'*Besides, a new borehole (JB-B-I) was drilled down to 60.6 m near the above-mentioned AWS in March 2017.*' according to your comments.

Line 116: Do you mean "wireless data transmission"? This also appears in line 157. Check!

Thanks. It was changed to

'*Using such technology, it would be possible to check collected data in real-time and identify possible sensor failures.*'.

Line 145: "…… other profiles were done using ……"
Line 147: "…… least-squares method was employed for ……"

Line 181: "…… protected by steel tube ……"

Line 187: Be careful to use "would be" herein, you can just say "…… which was mainly related to the local topography ……"

Line 193: Should say intra-permafrost groundwater.

Thanks for the comments, the above sentences have been revised as suggested and highlighted in red in the revised manuscript.

Figure 3. I noticed that there are "zero curtain" phenomena in the left panel (a1-a4) of Fig. 3, but the duration time of zero curtain of each curve is not the same. The authors should explain the possible reasons for this (I suppose this is mainly related to the in situ soil water/ice content of the permafrost sites).

The reason for different durations of zero curtain was added and highlighted in red in the revised manuscript.

'*Zero-curtain effects were evident at a particular depth in these five sites, but the duration time of zero curtains at the same depth varied greatly with location, which was mainly related to in situ soil water/ice content of these permafrost sites.*'.

Figure 6: To facilitate comparison, I suggest using the same range of values in the vertical axis (GT) for each subfigure. (Same for the left and right panels in Fig. 3)

We re-draw Figure 6 (revised as Figure 7 in the new version) according to your suggestion.

[Figure]

Figure 7. Variations in ground temperatures at the depth of 3 m on the right-of-way (ROW) at the JS (a), SL (b), and JB (c) sites along the China-Russia Crude Oil Pipeline (CRCOP) I in northern part of Northeast China during 2017–2021.

Line 228-229: This deepening of the permafrost table and warming of permafrost has exposed the pipelines to thawed low-bearing foundation soils. This sentence has two subjects, so you should use have instead of has.

Line 230: the CRCOP-I had locally settled down ……

Line 233: In the parenthesis, you can refer to the literature Zhang, T. (2011) when explaining "talik".

Zhang, T., 2011. Talik. In: Singh, V.P., Singh, P., Haritashya, U.K. (eds) Encyclopedia of Snow, Ice and Glaciers. Encyclopedia of Earth Sciences Series. Springer, Dordrecht. https://doi.org/10.1007/978-90-481-2642-2_563

The above sentences have been revised and relevant references have been added in the revised manuscript according to your comments.

Line 234-236: Why you used 300 ohm·m here? In my knowledge, this value can vary greatly from place to place around the world. The authors should give a criterion or any references to support their choice (e.g., Schön, J. H., 1996; Christof Kneisel et al., 2008). Also, you said "boundary", it is a little bit confusing. I understand you mean the boundary between unfrozen and frozen zones. So, my suggested wording is: electrical resistivity value of 300 ohm·m was used as the critical value to identify the boundary between frozen and unfrozen zones.

Schön, J. H. (1996). Physical properties of rocks. Handbook of geophysical exploration. Pergamon Press.

Christof Kneisel, Christian Hauck, Richard Fortier, & Brian Moorman, 2008. Advances in Geophysical Methods for Permafrost Investigations, Permafrost and Periglacial Processes, 19: 157–178, DOI: 10.1002/ppp.616

What you said is right. We selected a threshold value of electrical resistivity for discriminating between frozen and unfrozen zones according to the relationship between the resistivity, ground temperature, and lithology, as shown in the Figure below (added in the revised version and numbered Figure 12). The related description has been revised according to your comment.
'*Here, an ER value of 300 Ωm was used as the critical value to identify the boundary between frozen and unfrozen zones combined with the profile characteristics of resistivity, GT, water/ice content, and lithology (obtained from borehole drilling) (Fig.12).*'.

[Figure]

Figure 12. Dependence of electrical resistivity on ground temperature and lithology.

Line 254: Lower rate, not slower rate.
Line 256: Delete "the" before "cooling rate".
Line 257-258: "abnormal changes in 0 °C isotherm observed in Figures 9b, 9c, and 9d ……" Here, the using of "observed" is strange to me. Figures can never observe anything by itself. So, just say "showed in figures 9b, 9c ……"

Line 259: intra-permafrost groundwater.

Figure 10: Should give the unit for VWC, m3/m3 or %.

Line 287: The MAGT at 15 m depth……

Line 289: I suggest using "occurred" rather than "observed".

Line 292: the ERT results

We have checked and revised the above English writing problems in the manuscript according to your above comments.

**Reviewer 3:**

**3.1 General Comments**

This paper presents an in-situ observational dataset on permafrost thermal regimes along the China-Russia Crude Oil Pipeline (CRCOP) route in northeast China, consisting of meteorological observations, soil temperature and soil moisture content data, and electrical resistivity tomography (ERT) data. The analysis of this dataset shows that the operation of CRCOP has had profound effects on the thermal state of the ground. The results are useful for better understanding the responses of permafrost to climate change and engineering activities. This dataset is valuable for evaluating the integrity of pipelines and the effectiveness of measures to mitigate the permafrost thaw, as well as for validating numerical models. The manuscript is overall well organized and written, but there are still some shortcomings that need to be addressed.

**3.2 Specific Comments**

1, Please enhance the description of data processing (e.g. how do you deal with missing data and how these missing data affect the results?) in the revised manuscript. This will be important for full understanding of this dataset.

Thank you for your suggestion. We have added the description of quality control of data in the revised manuscript. Obvious erroneous recordings are manually removed and all the missing or abnormal data are replaced with null values before daily average values are calculated.

2, Warm-oil pipeline dissipates heat into the surrounding permafrost, resulting in thermal and physical disturbances to the pipeline right-of-way. These disturbances can compromise pipeline integrity and pose the potential risk of oil leakage. Therefore, in-situ permafrost monitoring has been made along several important pipeline routes (e.g. the Norman Wells pipeline and Trans-Alaska oil pipeline), where reliable first-hand data has been collected. I suggest the authors mention those important studies as background to this study.

Agreed. Pipelines constructed in permafrost are inevitably faced with major challenges related to thaw settlement, frost heave, slope failure, icing, and frost mounds. Therefore, the monitoring systems were established along several pipeline routes, such as the Norman Wells pipeline and the Trans-Alaska oil pipeline, to obtain field data for understanding how the permafrost foundation performed when a pipeline went through or above it. Such datasets and associated analyses are valuable because they can provide references and implications for CRCOP. According to your suggestions and comments from Reviewer 1, these important studies have been added to the introduction section in red in the revised manuscript as follows:

'*As a result, a permafrost monitoring network along the CRCOPs route was gradually established by referring to the experiences and lessons learned from other oil/ and gas pipelines (e.g. Norman Wells to Zama crude oil pipeline in Canada, Alyeska crude oil pipeline in the U.S., and Nadym–Pur–Taz natural gas pipeline in Russia) in permafrost regions (Burgess and Smith, 2003; Johnson and Hegdal, 2008; Smith and Riseborough, 2010; Oswell, 2011). Boreholes were instrumented to measure GTs in the active layer and near-surface permafrost on and off the right-of-way (ROW) of*

*the CRCOPs and electrical resistivity tomography (ERT) surveys were used to delineate frozen and unfrozen ground in the vicinity of the CRCOPs (Kneisel et al., 2008; Farzamian et al., 2020).'.*

Newly added references are as follows:

*Burgess M M, Smith S L. 17 years of thaw penetration and surface settlement observations in permafrost terrain along the Norman Wells pipeline, Northwest Territories, Canada. Proceedings of the Eighth International Conference on Permafrost, 2003: 107-112.*

*Johnson E R, Hegdal L A. Permafrost-related performance of the Trans-Alaska oil pipeline. Proceedings of Ninth International Conference on Permafrost, Fairbanks, AK, USA. 2008: 857-864.*

*Smith S L, Riseborough D W. Modelling the thermal response of permafrost terrain to right-of-way disturbance and climate warming. Cold Regions Science and Technology, 2010, 60(1): 92-103.*

*Oswell J M. Pipelines in permafrost: geotechnical issues and lessons. Canadian Geotechnical Journal, 2011, 48(9): 1412-1431.*

3, Section 2. I found the locations of the sites are not accurate enough, rounded to two decimal places. The locations of boreholes and ERT profiles are not given in the manuscript, nor in the associated dataset. I suggest the author can provide accurate locations (at least four decimal places in unit degrees) for these mentioned locations.

Thank you for your suggestions. We have provided accurate locations for the five monitoring sites in Table 1 with four decimal places in unit degrees. The boreholes and ERT profiles have been given ID according to where they exist. For example, for JB-B-1 the -B- indicates that is a 'borehole', and the prefix JB is an abbreviation for the site name. So, the locations of boreholes can be determined by their name prefixes.

4, Ground temperature was automatically collected by the dataloggers of RTB37a36V3 and CR3000 or measured manually with. Please provide a description of the errors occurring in these measurements.

Results show that the accuracy of the manual readings is estimated to be ±0.1 °C, which is lower than the RTB37a36V3 and CR3000 data loggers because the data measured with Fluke 87/89 was collected once at an instantaneous time (Juliussen et al., 2010). However, there are no overlapping data collected via these three methods at the same borehole, we can not quantify the deviation of the three different recording methods.

*Juliussen H, Christiansen H H, Strand G S, et al. NORPERM, the Norwegian permafrost database–a TSP NORWAY IPY legacy. Earth System Science Data, 2010, 2(2): 235-246.*

5, The ROW widths are equal at each monitoring site? Please give a clear description.

Thank you for this important comment. Generally, the ROW is approximately 20 m wide along the pipeline route except for some special sites, such as poor geological conditions. In the revised manuscript, the ROW width at each site has been added and the ROW edge was also added in the ERT images.

6, Figure 3, seems problematic in the caption. According to the caption, column (a) indicates the active layer, but ground temperatures in XT in (a2-4) were measured below zero degrees for

several consecutive years, which actually implies permafrost at this depth.

Thanks for pointing out this issue. We are sorry for this mistake. The caption was revised as

'*Variability of ground temperatures at depths of 0–3 m (a) and 8–20 m (b) at the undisturbed sites along the route of CRCOPs in Northeast China, 2018–2021.*'.

7, In Lines 110-111, please give exact timing for the borehole drilling.

The exact timing has been added in the manuscript, as described as

'*Besides, a new borehole (JB-B-I) was drilled down to 60.6 m near the above-mentioned AWS in March 2017.*'.

8, Line 232 to 240, ERT results show that a talik formed around CRCOP I is much larger than that around CRCOP II. What is the reason?

The CRCOP I was constructed during 2009-2010 and began to operate in 2011. To reach the requirement of 30-metric-million-ton throughput per year signed by the governments of China and Russia, an equal-size new pipeline, i.e., the CRCOP-II, in parallel with the CRCOP-I was built in 2017 and began operation in 2018. Due to a longer operation time and heating from the CRCOP I, the talik formed around CRCOP I is much larger than that around CRCOP II.

9, Line 255-257, it's difficult to directly observe the 1.5 m cooling range of two-phase closed thermosyphons and the 4 m lateral extent of thermal disturbance in Fig.9c and d, please clarify this point.

Thank you for your suggestion. Difficulties in understanding and even misunderstandings may arise if the distances of boreholes and thermosyphons from the pipe centerline were not provided. In the revised manuscript, this detailed information has been added in Table 3.

10, In Figure 8 and Appendix D, please use blue for higher resistivity and red for lower resistivity in ER images, like the color scheme in Figure 9. This is more intuitive for readers.

We re-draw Figure 8 (revised as Figure 11 in the new version) and Appendix D according to your suggestion. More detail can be found in Section 1.2 in this file.

11, In Figure 10, please add the unit for soil volumetric liquid water content.

Yes, we added the unit in Figure 10 in the revised version.

---

## Author Response (AR2)

**Response Letter**

The manuscript has been revised and improved again according to the comments raised by two reviewers and the Editor and highlighted in blue in this version of the manuscript. A list of responses is itemized as follows.

Sincerely,

Guoyu Li, Huijun Jin and Fei Wang

**Comment 1**

I am satisfied with the revision, which took into account most of my initial comments. The quality of the manuscript is much improved.
Other comments:
Lines 248 to 266 and figure 9: they are about the cooling effect of thermosyphons and thermal disturbance from the warm pipeline. If I am correct, these four boreholes are all affected by both the thermosyphons and pipeline. It would be more convincing if there was an undisturbed borehole as a control to distinguish the effect of the pipeline.

Yes. Four instrumented cross sections were established perpendicular to the CRCOP I at the JB permafrost site, as shown in Figure 3C (Appendix C). Ground temperatures in the boreholes of JB-B-1 (on-ROW, 2 m away from the pipe) and JB-B-II (off-ROW, 16.6 m away from the pipe) are used as the reference values to analyze the thermal disturbance of pipeline and cooling performance of the thermosyphons, which are analyzed and discussed in Section 3.2.1 and Section 3.2.2 in detail.

Figure 6, better to remove the shadows under the labels. In (b), I suggest using x in place of a in the equations. Otherwise, you need to explain what is the variable a.

Thanks. This figure has been revised according to your suggestions.

[Figure]

Figure 6. Variations in the active layer thickness (ALT) (a) and mean annual ground temperature (MAGT) (b) in borehole JB-B-II at the JB site along the China-Russia Crude Oil Pipeline (CRCOP) I in northern part of Northeast China from 2011 to 2020.

Figure 8, please explain k in the figure caption.

The 'k' in Fig.8 denotes the rates of the permafrost table depth increasing and the

supra-permafrost subaerial talik thickening. The sentence 'Where k denotes the increasing rate, has been added and highlighted in blue in the new manuscript.

Figure 9, the term "Time series of temperature contours" is weird. In many occasions it is called time contour plots or depth-time contour plots.

We are sorry for this vague statement. The caption has been revised as

'*Depth-time contour plots of ground temperature (°C), derived from the boreholes JB-B-2 (a), JB-B-3 (b), JB-B-6 (c), and JB-B-9 (d) at the JB site along the China-Russia Crude Oil Pipeline (CRCOP) I in the northern part of Northeast China. The blank gap indicates the missing data*'

**Comment 2**

The detailed response to review comments provided by the authors is much appreciated. Most of the comments have been adequately addressed. Additional information and clarifications have been provided that have improved the manuscript. A better description of the instrumentation and its location with relative to the pipeline ROW, has been provided to the reader. Additional minor revisions however are required for the paper to be acceptable for publication. Some of the comments are related to new text or figures included in the revised draft (see below). I have also provided a number of suggestions for editorial revisions that should also improve the paper.
Specific Comments
L49-50 – The development of taliks is part of shrinking permafrost extent. I suggest your rewrite the sentence: "…thickening active layer, development of taliks, shrinking permafrost extent and increases in thaw-related landscape change and hazards such as ground surface subsidence, settlement of foundation soils and thermokarst."

Thanks for your better suggestions. This sentence has been revised as suggested and highlighted in blue in the new manuscript.

L55-56 – This sentence seems out of place as the previous few sentences are commenting on expected change rather than current extent. You could consider moving this sentence to the beginning of the paragraph where you are discussing the distribution of permafrost in the region.

Thanks. This sentence has been moved to L44-45 and also highlighted in blue.

L63 – You should probably say something about when the pipelines were constructed and when operation started.

Thanks. The CRCOP I was constructed during the two cold seasons in 2009–2010 using a conventional burial construction method and became operational in January 2011. The CRCOP II, nearly parallel to CRCOP I, was built in 2017 and began operation in January 2018. The separation distance between the two parallel pipelines is generally limited to approximately 10 m. The two sentences have been added in L65-66 in the new version according to your comments.

'*The CRCOP I was constructed during the two cold seasons in 2009-2010 and began operation in January 2011. In January 2018, CRCOP II was completed and began to operate.*'.

L67 – Any references for Nadym-Pur-Taz pipeline?

The reference has been added.

*Seligman B J. Long-term variability of pipeline-permafrost interactions in north-west Siberia. Permafrost and Periglacial Processes, 2000, 11(1): 5-22.*

L92 – Is the permafrost thickness >60 m everywhere in the region? You might want to give a range, i.e. from Xm to more than 60 m thick.

Thanks. This sentence was revised as '*The permafrost thickness varies from 0 to more than 60 m, ...*', based on the published papers from Jin et al. (2007) and Wang et al. (2019).

*Jin H, Hao J, Chang X, et al. Zonation and assessment of frozen-ground conditions for engineering geology along the China–Russia crude oil pipeline route from Mo'he to Daqing, Northeastern China. Cold Regions Science and Technology, 2010, 64(3): 213-225.*
*Wang F, Li G, Ma W, et al. Pipeline–permafrost interaction monitoring system along the China–Russia crude oil pipeline. Engineering Geology, 2019, 254: 113-125*

L95 – replace "where" with "and"

It was changed.

L106 – Table 1 notes – specify if MAGT measured at DZAA

Thanks. It was revised as '*MAGT, mean annual ground temperature of permafrost at the depth of zero annual amplitude, ...*' according to the comment.

L118 – Table 2 – Accuracy and Resolution (Precision) are not the same thing so there should be two values provided.

Thanks for pointing out this issue. The accuracies for sensors are given in Table 2 and the word 'resolution' in Table 2 has been deleted.

L126-127 – Revision suggested: "….at ambient temperature of 20 °C, but only +/-0.34 °C at…"

It has been revised and highlighted in blue.

L137 – Do you also have to adjust the daytime readings – do the have the same systematic error?

In the processing of total solar radiation data measured by the LI200X pyranometer, the negative values are set to zero and other values keep invariant (refer to the instruction manual, https://www.campbellsci.com/li200x-l).

L151-152 – It isn't clear from Table 3 if JB-B-1 is on or off the ROW because it appears under both categories in the table.
L153-154 – SL-B-1 – located both on and off the ROW? Check borehole numbers.
L155-157 – Similar comment to above. You seem to be indicating that JS-B-1 is located both on and off the ROW.
L162 Table 3 – See previous comments. The same borehole seems to be identified as being both on and of the ROW. Be clear on location of sites in both the text and table.

The boreholes have been named according to where they exist. For example, for JB-B-1 the -B- indicates that is a 'borehole', and the prefix JB is an abbreviation for the site name. To distinguish

whether the borehole is on the pipeline ROW, Arabic numerals are used to name the boreholes on-ROW (e.g., JB-B-1) and the off-ROW boreholes are numbered using Roman numerals (e.g., JB-B-I). Thus, the locations of boreholes can be determined by their name prefixes and suffixes. For example, 'XA-B-I' indicates the borehole is located off the pipeline ROW at the XA site.

L174 – Delete "Besides"

It was deleted.

L184-185 – Delete "dynamic". Also revise to "..but the amplitude decreased with depth, with the magnitude of the decrease varying between sites."

Thanks. This sentence has been revised as you suggested.

L186-188 – You are confusing amplitude and range. The values given are the range in temperature. Note amplitude is measured from the mean value. Revise text to: "..maximum annual range of GT…….XT had the maximum range in GT….."

We are sorry for this serious mistake. Thanks for your comments. It was revised and highlighted in blue.

L189 – What is meant by "at a particular depth" – upper few metres?

It was an unhappy choice of words and revised as '*at a certain depth*'.

L192 – Figure 3 – Which JB site is being shown, JB-B-1 or JB-B-11? In my comments on the earlier draft, I had asked whether the variation in the deeper temperatures might be due to an issue with a sensor. For JB there appear to be spikes (sharp increases) in the temperature record. We see these periodically in our data and it is likely due to some electrical glitch with sensor and logger. This fluctuation of up to 0.5°C at depths of 10-20 m does not seem to show up in the profiles shown in Figure 5. For JS, the "noise" for the deeper sensors seems to be greater than that for the shallower sensors. The record also doesn't appear to be continuous and perhaps the authors have removed data points they determined were erroneous. If the temperatures measured by the deeper sensors are correct then it would seem that the temperature does not stay below 0°C throughout the year at any depth so perhaps no permafrost present?

Up to 38% of the temperature data in borehole JB-B-II were missing in 2019 (between late March and early August) due to the detachment of the thermistor cable from the CR3000 data logger. Therefore, the GTs measured in borehole JB-B-I are graphed in Fig.3, considering continuity and long time series of data. The sharp increase in the deeper temperatures might be due to some electrical glitch with the sensor and logger, as you stated. For assessment of the inter-annual trend of permafrost off the ROW, a decade record of GTs in borehole JB-B-II was used. Fig.5 shows the monthly average GTs in 2018 in this borehole. Therefore, abnormal fluctuation in temperatures at depths of 10-20 m does not show up in the temperature profiles.

At the JS site, the off-ROW borehole JS-B-I and two on-ROW boreholes of JS-B-1 and JS-B-2 are all connected to a data logger. For these two boreholes on-ROW, high-quality temperature data series were collected. Thus, the sharp increase in the deeper temperatures for the JS-B-I borehole in the summers of 2019 and 2020 was not caused by the data logger. The phenomenon of temperature increase at depths of 10-20 m was observed for two consecutive years (Fig.R1),

which would seem to indicate that the failure of the thermistors is impossible. As Fig. R2 show, Large surface runoff occurs in low-lying areas at this site in mid-June, resulting in occasional floods and waterlogging. Meanwhile, the strata of this site (obtained from the JS-B-I borehole) are composed of peaty soil (0-0.8 m), grey silty clay (0.8-2.0 m), yellowish-brown gravel (2.0-5.5 m), and strong weathered granite (5.5-20.0 m). The presence of the rock layers with high permeability provides a channel for the movement of supra- and/or intra-permafrost groundwater. For the above-mentioned reasons, we conclude the positive temperatures that occurred from late June to September are probably due to the thermal disturbance of supra- and/or intra-permafrost groundwater.

[Figure]

Figure R1. Monthly average ground temperatures at depths of 0-20 m recorded in the JS-B-I borehole at the JS site in 2019 (a) and 2020 (b).

[Figure]

Figure R2. Waterlogging on the ground surface at the JS site on June 17, 2021.

L195 – Revise to "Seasonal variations in GTs…."
L199-200 – Revision suggested: "…decreased northward. However there is substantial scatter in the relationship between GT and latitude (Fig. 4)."

They were revised based on the above suggestions.

L202 – Fig. 4 – Why is JS not included?

Thanks. As shown in Fig.R1 and R2, positive temperatures (at depths of 12 -20 m) were observed in the summers of 2019 and 2020, therefore, the data for the JS site was not graphed in Fig.4.

L204 – Delete "under a warming climate" (this refers more to attribution of the trend rather than

determining the direction and magnitude of the trend)

Thanks for your precious comments. It was deleted.

L205-206 – Do you mean the profile is isothermal? This has been shown for warm permafrost in other regions – see for example Smith et al (2010, DOI: 10.1002/ppp.690); Romanovsky et al. (2010, DOI: 10.1002/ppp.683)

Thanks for pointing out this issue. Figure3 shows that large annual variation in temperatures near the surface decreases to little amplitudes at the depth of 1 m of the ground. At depths from 1 to 2 m, the monthly average GTs in 2018 was fluctuating in proximity to 0 °C, which allow permafrost to persist for decades due to latent heat effects. These two references have been added in the revised manuscript.

L209 – This also depends on the thermal properties of the material.

According to the comment, this sentence has been revised as '*...layer and thermal properties of soil deposits.*'.

L223 – Revision suggested: "….buried at a depth of 1.6 – 2.4 m…."
L225 – Delete "ambient"
L228 – Delete "Besides"

They have been revised in the new manuscript.

L232-233 – The information on construction and start of operation of pipeline should have been mentioned earlier in the paper. Revision suggested: "…since initiation of CRCOP 1 operation in 2011,…."

According to the above comments, the related information has been added in L65-66 in the new manuscript. The sentence was also revised as you suggested and highlighted in blue.

L238-240 – If the ground is ice-rich there would also be significant settlement of the ground surface accompanying the increase in thaw depth as has been shown for other pipelines.

Agree. Significant ground surface subsidence within the trench of the pipeline occurred along the CRCOPs, particularly in warm and ice-rich permafrost zones. Up to 1 m of surface subsidence within the trench area had been measured just ten months after the initiation of CRCOP 1 operation (i.e., in October 2011) during the field surveys. So the sentence has been revised as '*...led to significant subsidence of the ground surface within the trench area and...*'.

L253-255 – In response to review comment regarding use of "artificial permafrost table", the authors indicated that this was revised to just refer to "permafrost table". However the term is still used here.

The term of "artificial permafrost table", less frequently used, is used to define the changing permafrost table under the influence of engineering activities (e.g., Wu et al., 2016). We are very sorry that this term had not been revised in this paragraph. In this revision, this term has been revised and its abbreviation in Appendix A is deleted.

*Wu Q, Zhang Z, Gao S, et al. Thermal impacts of engineering activities and vegetation layer on*

*permafrost in different alpine ecosystems of the Qinghai–Tibet Plateau, China[J]. The Cryosphere, 2016, 10(4): 1695-1706.*

L255 – Revision required: "Overall the depth of…." (need to be clear it is the depth of the permafrost table that is increasing).
L258 – Delete "Besides"

They have been revised.

L259-260 – The lateral extent of the thermal disturbance isn't clearly shown by these types of figures. A figure showing the temperature distribution across the ROW would be more effective, i.e. something comparable to the ERT figure.

As shown in Figure C3 (Appendix C), the boreholes of JB-B-3, JB-B-6, and JB-B-9 are located along the same cross-section of the pipeline, which are 2, 3, and 4 m away from the pipeline centerline, respectively. The increasing rate of the permafrost table depth in the JB-B-9 borehole is still greater than that of off-ROW borehole of JB-B-II over the observational period, suggesting the thermal disturbed range of the warm pipeline is greater than 4.0 m.

L263-264 – What is meant by unexpectedly warmer oil temperature? Was the temperature higher than the original design values?

Yes. The oil temperature at the first pump station of CRCOP during the preliminary design stage varied from −6 to 10 °C, and later it was changed to −3.65 to 6.41 °C. However, the oil temperature is all above 0 °C year-round, ranging from 0.4 to 28 °C. Besides, it shows a slow increase trend with operational time increases.

L263-266 – Sentence isn't clear. Are you referring to surface settlement and the resulting ponding of water that occurs with thawing of ice-rich permafrost? Given the high temperature of the pipe, how important is the effect of climate change over this relatively short time period? Isn't the low performance of the thermosyphons mostly due the significant heat source (the pipe) that is present throughout the year?

Thanks. The heat dissipated from the pipeline led to rapid permafrost thawing and thaw bulb development around the pipe. Following the permafrost thawing and the thaw bulb expanding, ground settlement within the trench of the pipeline developed gradually due to the thaw consolidation of permafrost layers. Then, rainfall, surface water, and groundwater accumulated within the sinking trench of the pipeline in summer times, which would infiltrate to the permafrost table along the longitudinal cracks developed at the two shoulders of the trench and accelerated thawing. This phenomenon is very common and serious in the permafrost wetlands along the CRCOPs route. The warm oil temperature and thermal erosion of surface water ponding on-ROW are the two main causes of permafrost thaw around the CRCOPs, and are also responsible for the unsatisfactory cooling effect of thermosyphons on the pipeline foundation soils. Based on the above-mentioned comments, the sentence has been revised as "*The unexpectedly warmer oil temperature, thermal erosion of surface water ponding on the ROW, and lowering thermosyphon performance are responsible for the unsatisfactory cooling effect of thermosyphons on the pipeline foundation soils.*".

L273 – "soil layers" could be deleted.

Deleted as suggested.

L273-274 – VWC could be given as a %

It was changed and Figure 10 has been revised.

[Figure]

Figure 10. Temporal history of soil volumetric liquid water content at depths of 0.5, 1.5, and 2.5 m at the JB site along the China-Russia Crude Oil Pipeline (CRCOP) I in the northern part of Northeast China during 2015-2021.

L274 – revision suggested: "…is less variable with an ….."
L284-286 – revision suggested: " ERT can be utilized to delineate frozen and unfrozen ground along the….."

Thanks, they were revised according to the above comments.

L286 – If surveys were done in April then maximum thaw would not have occurred yet.

Yes. This fieldwork was conducted when the snow on the ground surface was completely melted because the freezing depth of strata often reached its maximum in March-April in Northeast China.

L289 – insert "steel" between "stainless" and "electrodes"

Added as you suggested.

L291-292 – Repetition of information in previous paragraph – you could modify the text in the previous paragraph to mention inverted ERT results and reduce repetition.

Thanks for pointing out this issue. This sentence has been deleted and the associated sentence in the previous paragraph has been revised as '*The ER distribution within the subsurface can be visualized by ERT. The inverted ERT results provide a continuous transect of the characteristics of the active layer and near-surface permafrost to delineate the shape and size of talik (unfrozen ground in permafrost regions) or permafrost islands along the CRCOPs route (Zhang, 2011).*'.

L293-295 – Figure 12 appears to indicate that values >300  m aren't necessarily associated with frozen ground. Was the resistivity and ground temperature measured at the same time? The ERT surveys were done in April but the temperature profile in figure 11 would appear to be from later in the thaw season. Material type is also important. For example, Lewkowicz et al. (2011, DOI:

10.1002/ppp.703) noted that for some of their surveys in the Yukon, moderate resistivity values were associated with bedrock or gravel rather than being an indicator of frozen ground. It isn't clear if you have utilized information on lithology in the interpretation of ground thermal conditions in figure 11. Holloway and Lewkowicz (2019, Cold Regions Engineering 2019 & 8th Can. Permafrost conf. https://doi.org/10.1061/9780784482599) also observed high values of resistivity for unfrozen silt and peat. Some additional explanation regarding the basis for use of 300    m is probably required here.

Thanks for pointing out this issue. The ERs of some rock and soil deposits have a clear range at room temperatures, such as clay (0.1–10 Ωm), mudstone (10–100 Ωm), and basalt (100–$10^5$ Ωm). Moisture content, salt content, pore structure, soil texture, temperature, whether the pore water is frozen, and other factors all influence a geotechnical material's resistivity (Kneisel, 2006; Hauck, 2013; Lewkowicz et al., 2011). The significant difference in ER value of soils in the frozen and thawed states (e.g., ER value of ice-rich permafrost is typically 10-100 times higher than that of unfrozen soils) makes the ERT be used in permafrost surveys widely. For interpreting the ERT results and improving interpretation accuracy, borehole drilling and ground temperature data are commonly used. For example, based on the relationship among ER of soils, stratum lithology, and ground temperature in boreholes, an ER value of 500 Ωm was used as the critical value to interpret permafrost islands above the thawed zone (Li et al., 2021). In our study, according to the ERT detection in Northeast China made by previous scholars (Hu and Shan, 2016; Li et al., 2021) and profile characteristics in boreholes, an ER value of 300 Ωm was chosen as the critical value to identify the boundary between frozen and unfrozen zones. The sentence has been revised as '*Here, an ER value of 300 Ωm was used as the critical value to identify the boundary between frozen and unfrozen zones combined with the profile characteristics of resistivity, GT, water/ice content, lithology (obtained from borehole drilling, Fig.12), and other ERT surveys in Northeast China made by previous scholars (Hu and Shan, 2016; Li et al., 2021).*' and the related references have been added.

*Hauck C. New concepts in geophysical surveying and data interpretation for permafrost terrain. Permafrost and Periglacial Processes, 2013, 24(2): 131-137.*
*Hu Z, Shan W. Landslide investigations in the northwest section of the lesser Khingan range in China using combined HDR and GPR methods. Bulletin of Engineering Geology and the Environment, 2016, 75(2): 591-603.*
*Kneisel C. Assessment of subsurface lithology in mountain environments using 2D resistivity imaging. Geomorphology, 2006, 80(1-2): 32-44.*
*Lewkowicz A G, Etzelmüller B, Smith S L. Characteristics of discontinuous permafrost based on ground temperature measurements and electrical resistivity tomography, southern Yukon, Canada. Permafrost and Periglacial Processes, 2011, 22(4): 320-342.*
*Li X, Jin X, Wang X, et al. Investigation of permafrost engineering geological environment with electrical resistivity tomography: A case study along the China-Russia crude oil pipelines. Engineering Geology, 2021, 291: 106237.*

L297 – permafrost thaw around the pipe?

Yes, the buried pipeline running at a positive temperature (up to 28°C), acting as a heat source, led to the permafrost around it thawing.

L299-230 – The profile seems to indicate that near the surface, frozen conditions exist and therefore difficult to see effect of ROW clearing. Given that the profile was done early in the thaw season it will be difficult to determine the overall effect on depth of thaw. The paper really doesn't mention much about surface disturbance during construction. The thermal effect of the pipe would appear to be the main influence on the ground thermal regime.

Thanks. This sentence has been deleted according to the comment.

L305- Fig. 12 – See previous comments on relationship between resistivity and temperature. What site are the data in the figure from? Are there borehole logs similar to this for all boreholes drilled? It would be useful to include the lithology and other information on material characteristics in the database.

The profile of soil resistivity, ground temperature, and lithology in borehole JB-B-1 are plotted in Fig.12 for interpreting ERT inversion results in Fig.11. The stratigraphy information obtained from borehole drilling at these monitoring sites was introduced in a paper published in the journal Engineering Geology in detail (Wang et al., 2019). According to the comment, the lithology of strata was added to ER dataset.

L314 – Typo – "60.6 m deep"

Revised as you suggested.

L314-315 – According to Table 3, volumetric water contents only appears to be available for 3 boreholes at the JB site rather than for all sites as the sentence implies. Meteorological data are also only collected at the JB site. You should rewrite the sentence to indicate that only the ground temperatures and ERT span the range in latitudes given. You should also specify that ground temperature and moisture content data were collected beneath the ROW and in undisturbed terrain off the ROW.

L318-319 – Revision suggested: "Analysis of data compiled indicates permafrost conditions along the eastern flank… Mountains are controlled…"

This sentence has been revised as you suggested.

L322-324 – Are you referring to the undisturbed sites here or those on the ROW? You need to be clear.

Thanks for pointing out this issue. Sorry for the unclear statement. This sentence has been revised as '…*GT measurements off the ROW indicates…*'.

L326 – revision suggested: "…leading to talik formation to a maximum depth…"

Revised as you suggested.

L327 – Poor sentence – revision required.

The sentence has been changed to '*This permafrost disturbance is still expanding and can persist for decades.*'.

L328 – Revision suggested: "…..permafrost beneath the pipeline ROW cannot be prevented but

can be significantly reduced by installing insulation or thermosyphons."

Revised as you suggested.